# GRACE: GENERATIVE REPRESENTATION LEARNING VIA CONTRASTIVE POLICY OPTIMIZATION

**Jiashuo Sun**[1]  **Shixuan Liu**[2]  **Zhaochen Su**[3]  **Xianrui Zhong**[1]  **Pengcheng Jiang**[1]
**Bowen Jin**[1]  **Peiran Li**[4]  **Weijia Shi**[5]  **Jiawei Han**[1]

[1]University of Illinois Urbana–Champaign    [2]Australian National University
[3]Hong Kong University of Science and Technology    [4]University of Wisconsin–Madison
[5]University of Washington

## ABSTRACT

Prevailing methods for training Large Language Models (LLMs) as text encoders rely on contrastive losses that treat the model as a black-box function, discarding its generative and reasoning capabilities in favor of static embeddings. We introduce GRACE (Generative Representation Learning via Contrastive Policy Optimization), a novel framework that reimagines contrastive signals not as losses to be minimized, but as rewards that guide a generative policy. In GRACE, the LLM acts as a policy $\pi_\theta$ that produces explicit, human-interpretable rationales—structured natural language explanations of its semantic understanding. These rationales are then encoded into high-quality embeddings via mean pooling. Using policy gradient optimization, we train the model with a multi-component reward function that maximizes similarity between query–positive pairs and minimizes similarity with negatives. This transforms the LLM from an opaque encoder into an interpretable agent whose reasoning process is transparent and inspectable. On MTEB benchmark, GRACE yields broad cross-category gains: averaged over four backbones, the supervised setting improves overall score by 11.5% over base models, and the unsupervised variant adds 6.9%, while preserving general capabilities. This work treats contrastive objectives as rewards over rationales, unifying representation learning with generation to produce stronger embeddings and transparent decision traces. Our code is publicly available at https://github.com/GasolSun36/GRACE.

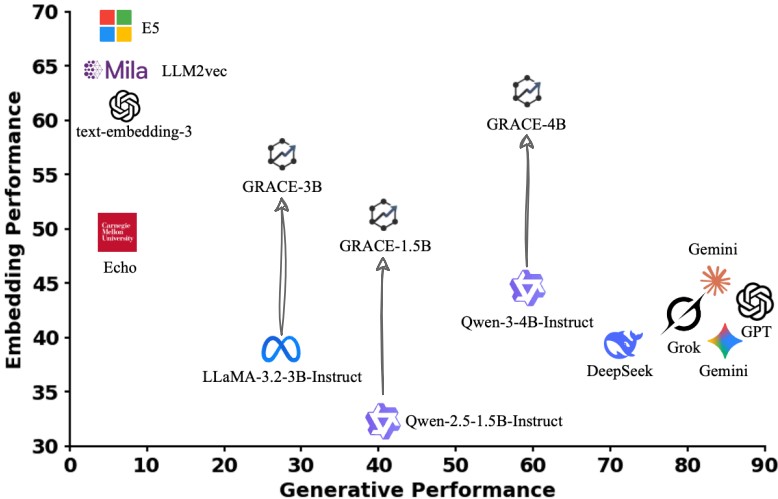

Figure 1: Joint comparison of generative and embedding performance across existing baselines and our GRACE models. Embedding models cluster at the top-left (strong embedding, weak generation), while generative models occupy the bottom-right (strong generation, weak embeddings). Trained without additional generative supervision, GRACE models shift instruction-tuned bases upward, simultaneously improving embedding performance while retaining generative competence.

# 1 INTRODUCTION

The advent of Large Language Models (LLMs) has marked a paradigm shift in the field of Natural Language Processing (NLP) (OpenAI, 2023; Yang et al., 2024; 2025; Dubey et al., 2024). Owing to their vast parameter scale and pre-training on massive text corpora, these models have demonstrated remarkable capabilities in language understanding, reasoning, and generation, leading to breakthroughs in nearly all NLP tasks. One of the most critical application areas is the use of LLMs as universal text encoders to provide high-quality semantic representations—or text embeddings—for downstream tasks such as semantic retrieval, text clustering, and recommendation systems Wang et al. (2024); BehnamGhader et al. (2024); Lee et al. (2025); Springer et al. (2025). From BERT Devlin et al. (2019) to contemporary instruction-tuned LLMs Ouyang et al. (2022), the research community has persistently explored methods to more effectively leverage the knowledge within these models to construct more accurate and robust representation spaces.

However, the prevailing paradigm for training LLMs as representation models harbors a profound inherent contradiction. Current approaches predominantly rely on contrastive learning frameworks that optimize discriminative objectives such as InfoNCE loss van den Oord et al. (2018), treating the LLM merely as a parameterized encoder function $f_\theta : \mathcal{X} \to \mathbb{R}^d$. This paradigm forces these inherently generative models to produce static embedding vectors through simple pooling mechanisms, fundamentally suppressing their capacity for structured reasoning and natural language generation. The model learns to minimize distances between positive pairs while maximizing distances from negatives, but this process occurs entirely within an opaque latent space. Consequently, we lose the interpretability that makes LLMs valuable—the ability to understand and articulate their reasoning process. When an embedding model determines that two texts are similar, we cannot inspect why it made that judgment or which semantic features it prioritized.

This fundamental limitation motivates us to reconceptualize the role of contrastive signals in representation learning. Rather than treating contrastive objectives as loss functions to be minimized through gradient descent, we view them as reward signals that guide a generative policy. This perspective naturally leads to a reinforcement learning framework in which the LLM acts as a policy $\pi_\theta$ that generates interpretable understandings of input texts. These understandings serve a dual purpose: they provide human-readable explanations of the model's semantic reasoning and are simultaneously encoded into high-quality representations of the inputs. By formulating representation learning as a sequential decision-making problem, we leverage the full generative capacity of LLMs, yielding interpretable reasoning and effective text representations.

To realize this vision, we present GRACE (Generative Representation Learning via Contrastive Policy Optimization), a framework that turns LLMs into interpretable representation learners using policy-gradient optimization. The model first produces explicit rationales $r$ that analysizes and reasoning the input. From $r$ we derive the final embedding $\mathbf{h}$ via mean pooling over hidden states. We recast contrastive learning signals as rewards that increase query–positive similarity and decrease query–negative similarity. Optimizing this reward with standard policy-gradient methods teaches the model to generate faithful rationales while simultaneously learning effective text representations.

The main contributions of this work can be summarized as follows. First, we present the first empirical evidence that rewards derived from contrastive learning can be leveraged to train policy models, resulting in improved representational capabilities. Second, we propose a novel methodology that enables the transformation of existing LLMs into powerful representation models while preserving their general-purpose capabilities without performance degradation, as shown in Figure 1. Third, this work represents a substantial advancement in text representation interpretability, as the model's reasoning can be directly inspected through its textual outputs. Fourth, our method yields a significant performance gain of avg 11.5 % over baseline models when evaluated on the MTEB benchmark. Finally, to facilitate reproducibility and advance future research in this domain, we will make all models, datasets, and code publicly available.

# 2 GENERATIVE REPRESENTATION LEARNING VIA CONTRASTIVE POLICY OPTIMIZATION

We propose to fundamentally reimagine the role of contrastive signals in representation learning. Rather than treating them as loss functions to be minimized, we reconceptualize them as reward

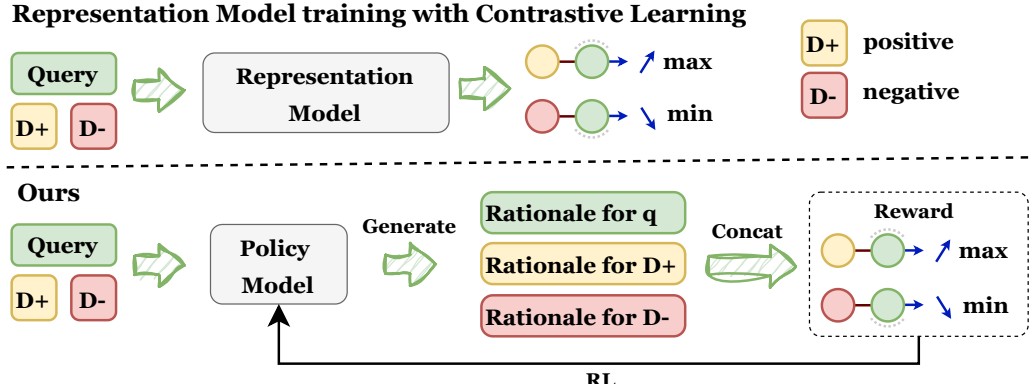

Figure 2: Comparison of standard contrastive learning (top) and our RL-based method (bottom). Given a query with positive ($D^+$) and negative ($D^-$) documents, our policy model generates rationales for $q$, $D^+$, and $D^-$, concatenates them to obtain the final representation, and is optimized with rewards that increase similarity between the $q$ and $D^+$ while decreasing similarity between the $q$ and $D^-$.

signals that guide a generative policy. Our framework transforms the LLM from a passive encoder that outputs static embeddings into an active agent that generates interpretable rationale of the input text.

## 2.1 RATIONALE-GENERATING POLICY

To enable the LLM to perform structured reasoning for similarity judgments, we adopt a policy that generates an explicit reasoning trace (rationale) for each input. Let $\mathcal{P}(\cdot)$ denote the prompting function that prepends the representation instruction to the input text. For any input $x \in \{q, d^+, d^-\}$, the policy $\pi_\theta$ produces a structured rationale:

$$r \sim \pi_\theta(\cdot \mid \mathcal{P}(x)). \tag{1}$$

The rationale $r$ identifies salient semantic features, key concepts, and potential relations grounded in domain knowledge, providing a transparent trace that supports downstream similarity assessment.

## 2.2 FROM RATIONALE TO REPRESENTATION

Given the generated rationale $r$ for input $x$, we obtain contextualized hidden states by conditioning on both the instruction-augmented input and the rationale:

$$\mathbf{E} = \pi_\theta(\mathcal{P}(x) \oplus r) \in \mathbb{R}^{L \times d}, \tag{2}$$

where $L$ is the sequence length, $d$ is the hidden dimension, and $\oplus$ denotes concatenation. To focus on semantic content while excluding instructional artifacts, we apply masked mean pooling over the last-layer hidden states:

$$\mathbf{h} = \frac{1}{|\mathcal{M}|} \sum_{t \in \mathcal{M}} \mathbf{E}_t, \quad \mathcal{M} = \{\, t : L_{\text{sys}} < t \leq L, \ \text{mask}_t = 1 \,\}, \tag{3}$$

where $\mathbf{h} \in \mathbb{R}^d$ is the final representation and $L_{\text{sys}}$ denotes the length of system prompt tokens to be excluded. Anchoring representation extraction to an explicit rationale yields semantically rich, interpretable embeddings and enables more reliable similarity judgments.

## 2.3 CONTRASTIVE REWARDS AS POLICY GUIDANCE

Given the recent advances of outcome-reward and policy optimization methods in LLMs, and the practical difficulties and inefficiencies of training value models, our algorithm is correspondingly designed around a policy-based optimization approach. For exposition, we adopt Group Relative Policy Optimization (GRPO) Shao et al. (2024) to walk through the procedure, though our framework is agnostic to the specific policy-gradient algorithm.

### 2.3.1 ROLLOUT STRATEGY

To balance exploration diversity with computational efficiency, we adopt an asymmetric rollout strategy. For each training instance $(q_i, d_i^+, d_i^-)$, we employ different generation strategies based on the text type:

$$y_{q_i} \sim \pi_\theta(\cdot|\mathcal{P}(q_i)), \quad y_{d_i^-} \sim \pi_\theta(\cdot|\mathcal{P}(d_i^-)), \quad \{y_{d_i^+}^{(k)}\}_{k=1}^K \sim \pi_\theta(\cdot|\mathcal{P}(d_i^+)) \tag{4}$$

For positive documents $d_i^+$, we perform $K$ stochastic rollouts to generate diverse rationales, enabling the model to explore different interpretational perspectives of the same content. In contrast, queries $q_i$ and negative documents $d_i^-$ undergo single-sample generation to produce reference representations, which serves as fixed anchors for reward computation. The computed similarity-based contrastive rewards facilitate advantage estimation within the GRPO framework, driving policy updates that enhance both the quality and diversity of text understanding across all input types.

### 2.3.2 REWARD DESIGN

We design a composite reward function that translates contrastive learning objectives into actionable policy guidance. For each positive document rollout $y_{d_i^+}^{(k)}$, we compute four synergistic reward components that collectively shape the learning dynamics.

Let $\mathcal{D}_i^- = \{d_{i,m}^-\}_{m=1}^{M_i}$ be the set of negatives for query $q_i$. The foundation is the contrastive learning reward $\mathcal{R}_{\text{CL}}$, which encourages semantic alignment between queries and relevant documents while penalizing spurious correlations with irrelevant ones:

$$\mathcal{R}_{\text{CL}}^{(i,k)} = \text{sim}(\mathbf{h}_{q_i}, \mathbf{h}_{d_i^+}^{(k)}) - \sum_{m=1}^{M_i} \text{sim}(\mathbf{h}_{q_i}, \mathbf{h}_{d_{i,m}^-}). \tag{5}$$

To ensure semantic coherence across multiple interpretations of the same document, we introduce the consistency reward, which encourages similar representations among concurrent rollouts:

$$\mathcal{R}_{\text{consist}}^{(i,k)} = \frac{1}{K-1} \sum_{\substack{j \neq k}}^K \text{sim}(\mathbf{h}_{d_i^+}^{(k)}, \mathbf{h}_{d_i^+}^{(j)}) \tag{6}$$

Most critically, we incorporate hard negative mining inspired by in-batch negative sampling strategies. For each query, we identify the most challenging distractors (positives from other training instances that exhibit spuriously high similarity). Let $B$ denote the batch size and $l \in \{1, \ldots, K\}$ index rollouts. We select, for each other instance $j$, the maximum similarity across its rollouts to obtain the hardest distractor for the current query:

$$\mathcal{R}_{\text{hard}}^{(i)} = -\frac{1}{B-1} \sum_{\substack{j=1 \\ j \neq i}}^B \max_{1 \leq l \leq K} \text{sim}(\mathbf{h}_{q_i}, \mathbf{h}_{d_j^+}^{(l)}). \tag{7}$$

The composite reward integrates these complementary objectives:

$$\mathcal{R}_{\text{total}}^{(i,k)} = \mathcal{R}_{\text{CL}}^{(i,k)} + \lambda_1 \mathcal{R}_{\text{consist}}^{(i,k)} + \lambda_2 \mathcal{R}_{\text{hard}}^{(i)} \tag{8}$$

where $\lambda_1$ and $\lambda_2$ are hyperparameters that balance the relative contributions of consistency preservation and hard negative discrimination.

To sharpen the reward distribution and stabilize training dynamics, we apply temperature scaling to the composite rewards:

$$\hat{\mathcal{R}}_{\text{total}}^{(i,k)} = \frac{\mathcal{R}_{\text{total}}^{(i,k)}}{\tau} \tag{9}$$

where $\tau$ is the reward temperature parameter that controls the sharpness of the advantage distribution.

## 2.4 POLICY OPTIMIZATION OBJECTIVE

Following the GRPO framework, we compute advantages relative to the group baseline but remove standard deviation:

$$A^{(i,k)} = \mathcal{R}_{\text{final}}^{(i,k)} - \frac{1}{K} \sum_{l=1}^{K} \mathcal{R}_{\text{final}}^{(i,l)} \tag{10}$$

The policy is then optimized to maximize the advantage-weighted likelihood:

$$\mathcal{L}_{\text{total}} = -\mathbb{E}_{(q,d^+,d^-)\sim\mathcal{D}} \left[ \sum_{i=1}^{B} \sum_{k=1}^{K} A^{(i,k)} \log \pi_\theta(y_{d_i^+}^{(k)} | \mathcal{P}(d_i^+)) \right] \tag{11}$$

Since our optimization is on-policy, we omit importance sampling here.

## 2.5 UNSUPERVISED LEARNING EXTENSION

Inspired by SimCSE's Gao et al. (2021) unsupervised paradigm, we extend our framework to settings where only raw text is available without explicit query-document pairs. The key insight is that different interpretations of the same text should maintain semantic coherence while being distinguishable from interpretations of different texts.

Given a batch of texts $\mathcal{B} = \{x_i\}_{i=1}^{B}$, we perform asymmetric rollouts for each text:

$$y_{x_i}^{\text{anchor}} \sim \pi_\theta(\cdot | \mathcal{P}(x_i)), \quad \{y_{x_i}^{(k)}\}_{k=1}^{K} \sim \pi_\theta(\cdot | \mathcal{P}(x_i)) \tag{12}$$

Here the anchor serves as the positive counterpart for its own rollouts, directly analogous to SimCSE's same sentence positive constructed via independent noise, while the $K$ rollouts probe diverse yet semantically consistent interpretations of $x_i$.

We instantiate the unsupervised reward with the self-alignment term between the anchor interpretation and each rollout of the same text. Given the anchor representation $h_{x_i}^{\text{anchor}}$ and a rollout $h_{x_i}^{(k)}$, the reward is

$$R_{\text{self}}^{(i,k)} = \text{sim}\left(h_{x_i}^{\text{anchor}}, h_{x_i}^{(k)}\right). \tag{13}$$

The remaining terms, within-text consistency across rollouts of the same $x_i$ and in-batch hard-negative mining against other texts in the batch—are identical to their definitions in Sec. 2.3.2. For brevity, we omit their formulas here. When enabled, the overall unsupervised objective is the same weighted combination as in Sec. 2.3.2.

# 3 EXPERIMENT

## 3.1 DATASETS AND EVALUATION METRICS

We conduct comprehensive evaluations using the Massive Text Embedding Benchmark (MTEB) Muennighoff et al. (2023), a standardized framework that covers 7 task categories and 56 datasets, spanning retrieval (Retr.), reranking (Rerank.), clustering (Clust.), pair classification (PairClass.), classification (Class.), semantic textual similarity (STS), and summarization (Summ.). For representation aggregation, we adopt mean pooling Reimers & Gurevych (2019), explicitly excluding instruction tokens to avoid instruction-specific artifacts.

## 3.2 BASELINES

**Supervised Baselines** We compare four variants in the supervised setting: (1) **Base (No Training)** uses the pre-trained instruction-tuned model without any representation-specific optimization and serves as the starting point; (2) **Base w/ Reasoning** applies our reasoning prompt to the same model but performs no further training, isolating the effect of reasoning-style outputs on embeddings; (3) **Contrastive Learning (CL)** fine-tunes the base model with a standard InfoNCE objective using in-batch negatives Chen et al. (2020), representing the predominant contrastive approach; and (4) **GRACE** introduces reward-guided policy optimization that explicitly aligns generative reasoning with representation quality.

| Categories → 
 # of datasets → | Retr. 
 15 | Rerank. 
 4 | Clust. 
 11 | PairClass. 
 3 | Class. 
 12 | STS 
 10 | Summ. 
 1 | Avg. 
 56 |
|---|---|---|---|---|---|---|---|---|
| **`Qwen2.5-1.5B-Instruct`** | | | | | | | | |
| Base | 22.15 | 29.32 | 25.44 | 36.18 | 35.77 | 44.11 | 26.32 | 30.33 |
| w/ reasoning | 24.83 | 32.45 | 27.88 | 39.20 | 39.42 | 47.26 | 26.78 | 32.92 |
| w/ CL training | 38.95 | 43.88 | 36.21 | 52.02 | 53.87 | 56.39 | 28.43 | 43.21 |
| GRACE | 40.44 | 46.95 | 39.55 | 54.84 | 55.36 | 59.42 | 30.41 | 45.48 |
| **`LLaMA-3.2-3B-Instruct`** | | | | | | | | |
| Base | 31.28 | 38.16 | 32.05 | 48.12 | 47.36 | 59.25 | 27.78 | 39.34 |
| w/ reasoning | 33.21 | 40.44 | 34.28 | 51.27 | 50.89 | 61.78 | 28.14 | 41.54 |
| w/ CL training | 42.42 | 47.35 | 39.92 | 58.66 | 58.15 | 65.55 | 28.63 | 47.39 |
| GRACE | 44.01 | 49.12 | 41.30 | 60.44 | 60.72 | 64.02 | 29.10 | 48.49 |
| **`Qwen2.5-3B-Instruct`** | | | | | | | | |
| Base | 37.38 | 44.16 | 36.85 | 53.72 | 53.36 | 66.15 | 26.26 | 44.12 |
| w/ reasoning | 39.42 | 46.55 | 38.82 | 56.61 | 57.23 | 68.22 | 28.55 | 46.59 |
| w/ CL training | 45.90 | 52.87 | 43.26 | 74.08 | 65.94 | 70.05 | 29.68 | 52.10 |
| GRACE | 49.42 | 54.85 | 44.73 | 79.64 | 68.25 | 74.65 | 30.10 | 54.74 |
| **`Qwen3-4B-Instruct-2507`** | | | | | | | | |
| Base | 37.42 | 48.16 | 38.55 | 55.33 | 54.87 | 66.02 | 29.44 | 45.49 |
| w/ reasoning | 38.91 | 49.72 | 40.76 | 57.20 | 55.41 | 68.35 | 29.62 | 46.87 |
| w/ CL training | 48.66 | 53.38 | 43.02 | 78.81 | 69.94 | 74.12 | 29.91 | 54.34 |
| GRACE | 52.11 | 55.85 | 45.24 | 82.94 | 71.02 | 77.38 | 30.46 | 56.64 |

Table 1: Supervised results on MTEB.

**Unsupervised Baselines** In the unsupervised setting, we report: (1) **Other Open Models**, including representative encoder baselines (e.g., BERT Devlin et al. (2019), RoBERTa Liu et al. (2019)) and recent LLM-embedding methods (e.g., LLM2Vec BehnamGhader et al. (2024), Echo Springer et al. (2025)); (2) **Base (No Training)**, the same instruction-tuned model used zero-shot without representation-specific training; (3) **SimCSE**, which fine-tunes with the unsupervised SimCSE objective based on dropout-induced positives Gao et al. (2021); and (4) **GRACE**, which applies the same reward-guided policy optimization to align generative reasoning with embedding quality in an unsupervised regime.

**Implementation Details** We conduct experiments with four decoder-only language models: Qwen2.5-1.5B/3B-Instruct Yang et al. (2024), Qwen3-4B Yang et al. (2025), and LLaMA-3.2-3B-Instruct Dubey et al. (2024). For supervised training, we use a replication of the public portion of the E5 dataset Wang et al. (2024), which contains 1.5M samples following BehnamGhader et al. (2024). For unsupervised training, we follow the SimCSE Gao et al. (2021) setup without collecting any additional unlabeled corpus. All experiments are run on a single node with 4× NVIDIA H100 GPUs (94GB each). Training is performed for 2 epochs with a batch size of 64. We set the maximum prompt length to 1024 tokens and the maximum response length to 2048 tokens, applying right-side truncation when sequences exceed these limits.

## 3.3 MAIN RESULTS

**Supervised Results** Table 1 reports supervised MTEB results across seven task families for four decoder-only backbones and four training settings. Averaged across the four backbones, our method improves the MTEB average by 11.52% over the Base model, indicating consistent improvements across model sizes and families. The gains are broad based, with especially strong uplift on retrieval and pair classification, while classification, clustering, STS, and summarization also improve. The intermediate variants also contribute: explicit reasoning provides a modest but reliable increase, and contrastive training further enlarges margins; the final method delivers the most balanced cross-task performance.

| Categories → # of datasets → | Retr. 15 | Rerank. 4 | Clust. 11 | PairClass. 3 | Class. 12 | STS 10 | Summ. 1 | Avg. 56 |
|---|---|---|---|---|---|---|---|---|
| **Other Open Models** | | | | | | | | |
| BERT | 10.59 | 43.44 | 30.12 | 56.33 | 61.66 | 54.36 | 29.82 | 38.33 |
| RoBERTa | 62.63 | 29.05 | 56.95 | 41.92 | 8.62 | 55.24 | 28.64 | 37.86 |
| LLM2Vec$_{\text{LLaMA-3-8B}}$ | 24.75 | 49.20 | 39.74 | 65.91 | 69.00 | 67.85 | 25.59 | 48.84 |
| Echo$_{\text{Mistral-7B}}$ | 71.63 | 33.51 | 72.31 | 47.43 | 22.85 | 73.64 | 31.02 | 49.02 |
| **Qwen2.5-1.5B-Instruct** | | | | | | | | |
| Base | 22.15 | 29.32 | 25.44 | 36.18 | 35.77 | 44.11 | 26.32 | 30.33 |
| w/ SimCSE | 31.28 | 38.94 | 33.12 | 50.67 | 47.53 | 58.21 | 28.34 | 39.65 |
| GRACE | 34.57 | 41.86 | 35.49 | 51.12 | 50.78 | 56.44 | 29.07 | 41.45 |
| **LLaMA-3.2-3B-Instruct** | | | | | | | | |
| Base | 31.28 | 38.16 | 32.05 | 48.12 | 47.36 | 59.25 | 27.78 | 39.34 |
| w/ SimCSE | 35.42 | 41.27 | 34.96 | 52.88 | 50.73 | 65.44 | 29.24 | 43.00 |
| GRACE | 36.55 | 43.15 | 36.88 | 55.27 | 53.62 | 63.18 | 29.52 | 44.04 |
| **Qwen2.5-3B-Instruct** | | | | | | | | |
| Base | 37.38 | 44.16 | 36.85 | 53.72 | 53.36 | 66.15 | 26.26 | 44.12 |
| w/ SimCSE | 42.25 | 48.33 | 39.87 | 68.22 | 60.15 | 70.84 | 29.56 | 49.17 |
| GRACE | 43.15 | 49.72 | 41.58 | 70.44 | 62.91 | 69.38 | 29.63 | 50.15 |
| **Qwen3-4B-Instruct-2507** | | | | | | | | |
| Base | 37.42 | 48.16 | 38.55 | 55.33 | 54.87 | 66.02 | 29.44 | 45.49 |
| w/ SimCSE | 42.18 | 50.72 | 41.63 | 69.14 | 61.25 | 72.48 | 29.62 | 50.11 |
| GRACE | 43.67 | 52.34 | 42.87 | 70.05 | 62.73 | 71.66 | 30.16 | 51.03 |

Table 2: UnSupervised results on MTEB.

**Unsupervised Results** Table 2 summarizes unsupervised MTEB performance and shows a consistent stepwise improvement from Base, SimCSE and GRACE across all four backbones. Averaged across backbones, our unsupervised method improves the MTEB average by 6.85% over the Base model. Relative to widely used open baselines, our best unsupervised results (e.g., Qwen3-4B and Qwen2.5-3B) past LLM2Vec and Echo, while comfortably exceeding encoder-only BERT/RoBERTa averages. Improvements are broad based, with retrieval and pair classification benefiting most, and STS remaining strong without sacrificing performance on classification or clustering. These trends suggest that even without supervised signals, injecting reasoning-aware contrastive objectives yields robust, transferable enhancements over both naive LLM pooling and standard SimCSE-style tuning.

## 3.4 Ablation Studies

The MTEB benchmark covers a wide range of embedding tasks across diverse types, domains, and difficulty levels. For computational efficiency and comparability, following BehnamGhader et al. (2024), we evaluate a representative 16-task subset for ablations and analysis (Table A.5.1).

### 3.4.1 Ablation Analysis of Reward Function Design

We conduct a comprehensive ablation study on the two key hyperparameters in our reward function: consistency ($\lambda_1$) and hard negative mining ($\lambda_2$), evaluating a $5 \times 5$ grid of configurations with five discrete values for each parameter (0.0, 0.3, 0.5, 0.7, 1.0). Figure 3 presents the performance landscape for GRACE-3B under both supervised and unsupervised paradigms, revealing several critical insights. The results demonstrate that removing all reward constraints ($\lambda_1 = 0, \lambda_2 = 0$) yields poor performance for supervised and unsupervised training respectively, confirming the necessity of these structured reward signals. When only one reward component is active, pure hard negative mining ($\lambda_1 = 0, \lambda_2 = 1$) reaches 50.2 % and 45.2 %, while pure consistency weighting ($\lambda_1 = 1, \lambda_2 = 0$) achieves 49.3 % and 45.0 %, indicating the former provides a better baseline. Notably, the model exhibits significantly higher sensitivity to the hard negative mining weight ($\lambda_2$) compared to the

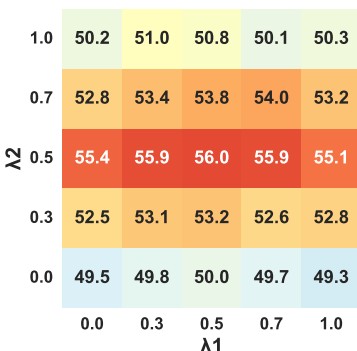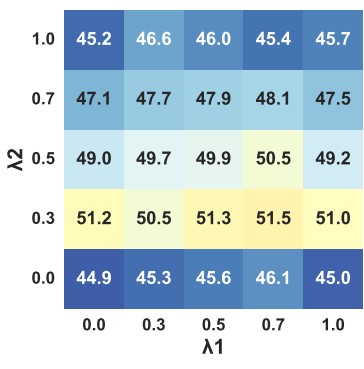

Figure 3: Reward function ablation study for GRACE-3B showing performance across different combinations of the consistency weight ($\lambda_1$) and the hard negative mining weight ($\lambda_2$). Left: supervised training; Right: unsupervised training. The heat intensity indicates performance levels, with darker red representing higher scores.

consistency weight ($\lambda_1$), suggesting that hard negative discrimination plays a more critical role in determining overall performance.

### 3.4.2 COMPARISON WITH ALTERNATIVE RL ALGORITHMS

To verify that our approach is not tied to a specific optimization scheme, we also applied it to different RL algorithms. As shown in Table 3.4.2, our method consistently improves performance across all three algorithms, demonstrating both its portability and generalizability. Among them, GRPO remains the most effective in our setting, while REINFORCE++, ReMax and DAPO bring smaller gains, likely because their design focuses on issues (e.g., reward hacking, weak credit assignment and long-CoT instability) that are less critical in our tasks. These findings confirm that our framework can be readily integrated with diverse RL optimizers, ensuring wide applicability.

| Algorithm
# of datasets → | Retr.
3 | Rerank.
2 | Clust.
3 | PairClass.
1 | Class.
3 | STS
3 | Summ.
1 | Avg.
16 |
|---|---|---|---|---|---|---|---|---|
| GRACE-3B | | | | | | | | |
| w/ ReMax Li et al. (2024) | 42.63 | 61.24 | 35.61 | 68.3 | 62.3 | 70.8 | 29.04 | 53.36 |
| w/ REINFORCE++ Hu et al. (2025) | 43.10 | 64.9 | 37.67 | 68.0 | 63.18 | 71.60 | 29.91 | 54.64 |
| w/ DAPO Yu et al. (2025) | 44.58 | 65.62 | 39.24 | 70.67 | 63.58 | 72.8 | 30.04 | 55.78 |
| w/ GRPO Shao et al. (2024) | 44.72 | 65.71 | 38.99 | 70.87 | 64.35 | 72.53 | 30.09 | 55.89 |

Table 3: Comparison of our supervised method with different RL algorithms on subsets of MTEB.

### 3.4.3 GENERALIZATION TO GENERAL DOMAIN TASKS

Table 4 evaluates whether embedding-oriented fine-tuning affects broad capabilities across mathematics (GSM8K Cobbe et al. (2021)), knowledge and reasoning (MMLU Hendrycks et al. (2021), BBH Suzgun et al. (2023), TriviaQA Joshi et al. (2017), FEVER Thorne et al. (2018)), and code generation (HumanEval Chen et al. (2021)). Across four backbones, our method preserves general performance: both supervised and unsupervised settings yield near-zero average shifts relative to the instruction-tuned base. In sharp contrast, the CL fine-tuning baseline suffers severe deterioration, indicating that naive contrastive objectives can substantially erode general-domain competence. We attribute the stability of our approach to the way the contrastive signal is integrated into learning. Rather than minimizing a token-agnostic loss, we optimize a contrastive reward within an RL framework, which (1) aligns updates with the generative policy, preserving instruction following and problem solving while shaping the embedding geometry; (2) uses relative, advantage-weighted updates that resist representation collapse and the rescaling/drift common in direct InfoNCE-style training; and (3) keeps reasoning generation intact so the policy continues to practice skills needed for general tasks. As a result, representations improve for retrieval objectives without sacrificing the general capabilities conferred by pretraining and instruction tuning.

Table 4: Performance on General Domain Tasks

| Dataset → | GSM8K | MMLU | TriviaQA | FEVER | BBH | HumanEval | Avg. | Δ |
|---|---|---|---|---|---|---|---|---|
| Metric → | EM | EM | EM | Acc | EM | Pass@1 | | |
| **Qwen-2.5-1.5B-Instruct** | | | | | | | | |
| Base | 32.06 | 54.94 | 18.35 | 66.91 | 25.25 | 46.95 | 40.74 | – |
| Bsse w/ CL training | 0.0 | 0.0 | 0.0 | 50.28 | 0.0 | 0.0 | 8.38 | **-32.36** |
| GRACE (Supervised) | 32.54 | 55.12 | 18.10 | 67.43 | 25.01 | 47.30 | 41.08 | **+0.34** |
| GRACE (Unsupervised) | 32.21 | 54.81 | 18.29 | 66.75 | 25.34 | 47.05 | 40.88 | **+0.14** |
| **LLaMA-3.2-3B-Instruct** | | | | | | | | |
| Base | 16.75 | 14.74 | 29.21 | 64.52 | 9.72 | 38.41 | 28.89 | – |
| Bsse w/ CL training | 0.0 | 0.0 | 0.0 | 50.01 | 0.0 | 0.0 | 8.33 | **-20.56** |
| GRACE (Supervised) | 17.02 | 15.01 | 29.05 | 65.10 | 9.61 | 38.90 | 29.27 | **+0.38** |
| GRACE (Unsupervised) | 16.81 | 14.69 | 29.18 | 64.40 | 9.80 | 38.55 | 28.91 | **+0.02** |
| **Qwen-2.5-3B-Instruct** | | | | | | | | |
| Base | 57.90 | 62.60 | 28.80 | 71.50 | 35.00 | 52.80 | 51.40 | – |
| Bsse w/ CL training | 0.0 | 0.0 | 0.0 | 50.13 | 0.0 | 0.0 | 8.35 | **-43.05** |
| GRACE (Supervised) | 59.10 | 61.20 | 27.60 | 72.80 | 34.30 | 53.10 | 51.50 | **+0.10** |
| GRACE (Unsupervised) | 58.00 | 61.50 | 28.30 | 71.20 | 34.80 | 52.90 | 51.10 | **-0.30** |
| **Qwen-3-4B-Instruct-2507** | | | | | | | | |
| Base | 75.96 | 69.45 | 31.55 | 83.53 | 35.01 | 68.90 | 60.73 | – |
| Bsse w/ CL training | 0.0 | 0.0 | 0.0 | 51.07 | 0.0 | 0.0 | 8.51 | **-52.22** |
| GRACE (Supervised) | 76.42 | 69.71 | 31.40 | 84.02 | 34.88 | 69.35 | 61.13 | **+0.40** |
| GRACE (Unsupervised) | 76.05 | 69.38 | 31.52 | 83.40 | 35.09 | 69.01 | 60.74 | **+0.01** |

## 4 RELATED WORK

**LLM as Embedding**   In recent years, the rise of large language models (LLMs) has sparked growing interest in using them directly as text embedding models Zhang et al. (2025); Yan et al. (2025); Ji et al. (2025). Research in this area has generally followed two paths: tuning-free approaches Li & Zhou (2025); Springer et al. (2025), which study the impact of instructions on embedding quality, and tuning-based approaches Muennighoff et al. (2025); BehnamGhader et al. (2024), which adapt models for improved performance. While tuning-free methods offer simplicity, state-of-the-art results increasingly rely on tuning-based strategies. Notable examples include the BGE Xiao et al. (2024) and GTE Li et al. (2023) series, which strengthen semantic representations through contrastive learning on large-scale text pairs. Building on this line of work, we reinterpret contrastive learning as a reward signal guiding a generative policy, turning LLMs into interpretable representation models.

**Reinforcement Learning in Reasoning**   Reinforcement learning has been central to advancing reasoning language models, exemplified by DeepSeek-R1 DeepSeek-AI et al. (2025), which achieved major breakthroughs with Reinforcement Learning with Verifiable Rewards (RLVR). RL formulates reasoning as a policy optimization, improving capabilities by maximizing expected rewards. Algorithms such as PPO Schulman et al. (2017) and GRPO Shao et al. (2024) dominate, with GRPO boosting efficiency by removing the critic model and adopting group-wise reward normalization. Extensions including DAPO Yu et al. (2025), ReMax Li et al. (2024), and REINFORCE++ Hu et al. (2025) further refine this paradigm. Inspired by these advances, we reinterpret contrastive learning as reward signal that guides a generative policy, thereby turning LLMs into representation models.

## 5 CONCLUSION

We propose GRACE (Generative Representation Learning via Contrastive Policy Optimization), a novel framework that reframes contrastive signals as rewards for a generative policy, turning LLMs from opaque encoders into interpretable representation learners. Optimized with standard policy-gradient updates, GRACE directly shapes the reasoning that gives rise to embeddings, yielding representations whose semantics are inspectable through the rationales. Empirically, GRACE delivers consistent, cross-category gains on MTEB across multiple backbones in both supervised and unsupervised settings, while preserving general capabilities on non-embedding tasks.

## 6 ETHICS STATEMENT

This work adheres to the ICLR Code of Ethics. All datasets used in our experiments, including the public portions of E5 and the MTEB benchmark, are publicly available and widely adopted in the community. No personally identifiable, sensitive, or proprietary data were collected or used. Our study focuses on methodological contributions to representation learning, without direct experimentation on human subjects or deployment in safety-critical applications. While we do not anticipate immediate societal harms, we acknowledge that improved embedding models could inadvertently amplify biases present in training data. We encourage future work to systematically evaluate fairness, bias, and robustness in broader contexts. The authors declare that there are no conflicts of interest related to this submission.

## 7 REPRODUCIBILITY STATEMENT

We have taken several steps to ensure the reproducibility of our work. A detailed description of our framework, including the reasoning policy architecture, rollout strategies, and multi-dimensional reward design, is provided in Sections 2 and 3. Comprehensive implementation details, such as model backbones, datasets, training configurations, and evaluation metrics, are reported in Section 3 and further expanded in the Appendix. To facilitate replication, we followed standardized benchmarks (MTEB) with publicly available datasets and describe all preprocessing steps in the supplementary materials. Our ablation studies 3.4 further validate the robustness of our design choices. We commit to releasing the codebase, trained model checkpoints, and scripts for data processing and evaluation in an open-source repository as soon as possible, so that all reported results can be independently verified.

## 8 ACKNOWLEDGEMENT

Research was supported in part by the AI Institute for Molecular Discovery, Synthetic Strategy, and Manufacturing: Molecule Maker Lab Institute (MMLI), funded by U.S. National Science Foundation under Award 2505932, NSF IIS 25-37827, and the Institute for Geospatial Understanding through an Integrative Discovery Environment (I-GUIDE) by NSF under Award No. 2118329. The research has used the Delta/DeltaAI advanced computing and data resource, supported in part by the University of Illinois Urbana-Champaign and through allocation #250851 from the Advanced Cyberinfrastructure Coordination Ecosystem: Services & Support (ACCESS) program, which is supported by National Science Foundation grants OAC 2320345, #2138259, #2138286, #2138307, #2137603, and #2138296.

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

# A APPENDIX

## A.1 THE USE OF LARGE LANGUAGE MODELS

In preparing this paper, we used LLMs solely as auxiliary tools to refine the clarity and style of our writing. Specifically, LLMs were employed to help polish phrasing and improve readability of paragraphs. All research ideas, experimental designs, analyses, and conclusions were conceived and executed by the authors. We take full responsibility for the content of this work.

## A.2 PRELIMINARIES

### A.2.1 CONTRASTIVE LEARNING FOR TEXT EMBEDDINGS

Traditional contrastive learning for text representation follows a discriminative paradigm. Given a dataset $\mathcal{D} = \{(q_i, d_i^+, \mathcal{D}_i^-)\}_{i=1}^N$ where $q_i$ denotes a query, $d_i^+$ denotes a relevant document, and $\mathcal{D}_i^- = \{d_{i,j}^-\}_{j=1}^m$ denotes irrelevant documents, an encoder $f_\theta : \mathcal{X} \to \mathbb{R}^d$ is trained to minimize the InfoNCE loss:

$$\mathcal{L}_{\text{InfoNCE}} = -\log \frac{\exp(\text{sim}(f_\theta(q_i), f_\theta(d_i^+))/\tau)}{\exp(\text{sim}(f_\theta(q_i), f_\theta(d_i^+))/\tau) + \sum_{d^- \in \mathcal{D}_i^-} \exp(\text{sim}(f_\theta(q_i), f_\theta(d^-))/\tau)} \quad (14)$$

where $\text{sim}(\cdot, \cdot)$ denotes cosine similarity and $\tau$ is a temperature parameter. In practice, in-batch negatives Karpukhin et al. (2020) are commonly employed for computational efficiency:

$$\mathcal{L}_{\text{batch}} = -\frac{1}{B} \sum_{i=1}^B \log \frac{\exp(\text{sim}(\mathbf{q}_i, \mathbf{d}_i^+)/\tau)}{\sum_{j=1}^B \exp(\text{sim}(\mathbf{q}_i, \mathbf{d}_j^+)/\tau)} \quad (15)$$

where $B$ is the batch size. The effectiveness of this approach critically depends on hard negative mining—identifying the most challenging negative samples that lie close to the decision boundary.

### A.2.2 POLICY GRADIENT OPTIMIZATION

Policy gradient methods optimize a parametrized policy $\pi_\theta$ by directly maximizing the expected reward over generated trajectories. Given a prompt $x$ and a policy $\pi_\theta$, the probability of generating a sequence $y$ is

$$\pi_\theta(y|x) = \prod_{t=1}^n \pi_\theta(y_t \mid x, y_{<t}). \quad (16)$$

With a reward function $r(x, y)$ that evaluates the quality of response $y$ to prompt $x$, the optimization objective is the expected reward:

$$J(\pi_\theta) = \mathbb{E}_{y \sim \pi_\theta(\cdot|x)} \big[ r(x, y) \big]. \quad (17)$$

By applying the policy gradient theorem, the gradient of the objective can be expressed as

$$\nabla_\theta J(\pi_\theta) = \mathbb{E}_{y \sim \pi_\theta(\cdot|x)} \left[ r(x, y) \, \nabla_\theta \log \pi_\theta(y|x) \right], \quad (18)$$

which provides an unbiased estimator of the true gradient Williams (1992); Sutton et al. (1999). To reduce the variance of policy gradient estimates, it is common to subtract a baseline $b(x)$ that does not depend on the sampled action. The gradient then becomes:

$$\nabla_\theta J(\pi_\theta) = \mathbb{E}_{y \sim \pi_\theta(\cdot|x)} \big[ \left( r(x, y) - b(x) \right) \nabla_\theta \log \pi_\theta(y|x) \big] \quad (19)$$

### A.2.3 GROUP RELATIVE POLICY OPTIMIZATION

Group Relative Policy Optimization (GRPO) treats model training as a reinforcement learning problem Shao et al. (2024). Given a policy $\pi_\theta$, GRPO samples a group of $G$ responses $\{y_1, ..., y_G\} \sim \pi_{\theta_{\text{old}}}(\cdot|x)$ from the old policy $\theta_{\text{old}}$ for each input $x$, and computes the advantage of each response relative to the group:

$$\hat{A}_i = \frac{r(x, y_i) - \frac{1}{G}\sum_{j=1}^{G} r(x, y_j)}{\text{std}(\{r(x, y_j)\}_{j=1}^{G})} \tag{20}$$

where $r(x, y_i)$ is the reward of $y_i$, and the token-level advantage $\hat{A}_{i,t} = \hat{A}_i$. The importance ratio is defined as $r_{i,t}(\theta) = \frac{\pi_\theta(y_{i,t}|x, y_{i,<t})}{\pi_{\theta_{\text{old}}}(y_{i,t}|x, y_{i,<t})}$. The objective without KL divergence is:

$$\mathcal{J}_{\text{GRPO}}(\theta) = \mathbb{E}\left[\frac{1}{G}\sum_{i=1}^{G}\frac{1}{|y_i|}\sum_{t=1}^{|y_i|}\min\left(r_{i,t}(\theta)\hat{A}_{i,t}, \text{clip}(r_{i,t}(\theta), 1-\varepsilon, 1+\varepsilon)\hat{A}_{i,t}\right)\right] \tag{21}$$

### A.3 TRAINING ALGORITHM

### A.4 THEORETICAL PERSPECTIVE ANALYSIS

Our framework establishes a principled connection between contrastive learning and reinforcement learning, leveraging their shared ability to learn from feedback rather than absolute ground truth labels. This fundamental similarity enables a natural integration of their objectives.

### A.4.1 UNIFIED LEARNING WITHOUT GROUND TRUTH

Both contrastive learning and reinforcement learning operate on comparative signals: contrastive learning distinguishes positive from negative samples, while reinforcement learning optimizes based on rewards signals. This parallel allows us to reformulate the contrastive objective as a reward signal:

$$\mathcal{L}_{\text{CL}} = -\log\frac{\exp(\text{sim}(q, d^+))}{\sum_{d'\in\mathcal{D}}\exp(\text{sim}(q, d'))} \quad\Rightarrow\quad \mathcal{R} = \text{sim}(q, d^+) - \sum_{d^-\in\mathcal{D}^-}\text{sim}(q, d^-) \tag{22}$$

This transformation enables policy gradient optimization without requiring explicit labels, only relative preferences encoded in the contrastive structure.

### A.4.2 CONNECTION TO INFONCE

Our framework can be understood as implicitly optimizing a generative variant of the InfoNCE objective. Consider the expected reward under our multi-faceted design:

$$\mathbb{E}_{\pi_\theta}[\mathcal{R}_{\text{total}}^{(i,k)}] = \mathbb{E}\left[\mathcal{R}_{\text{CL}}^{(i,k)} + \lambda_1\mathcal{R}_{\text{consist}}^{(i,k)} + \lambda_2\mathcal{R}_{\text{hard}}^{(i)}\right] \tag{23}$$

Expanding the contrastive learning component:

$$\mathcal{R}_{\text{CL}}^{(i,k)} = \log\frac{p(y_{d_i^+}^{(k)}|q_i, \pi_\theta)}{p(y_{d_i^-}|q_i, \pi_\theta)} \tag{24}$$

With hard negative mining across the batch:

$$\mathcal{R}_{\text{hard}}^{(i)} = -\log\left(\frac{1}{B-1}\sum_{j\neq i}^{B}\max_l p(y_{d_j^+}^{(l)}|q_i, \pi_\theta)\right) \tag{25}$$

Combining these terms, the expected total reward approximates:

$$\mathbb{E}[\mathcal{R}_{\text{total}}] \propto \log\frac{p(d^+|q, \pi_\theta)}{p(d^-|q, \pi_\theta)\cdot\prod_{j\neq i}\max_l p(y_{d_j^+}^{(l)}|q_i, \pi_\theta)} + \lambda_1\mathbb{E}[\text{consistency}] \tag{26}$$

---

**Algorithm 1** GRACE: Generative Representation Learning via Contrastive Policy Optimization

---

**Require:** Training data $\mathcal{D} = \{(q_i, d_i^+, d_i^-)\}_{i=1}^N$, Initial policy $\pi_\theta$
**Require:** Hyperparameters: rollouts $K$, batch size $B$, coefficients $\lambda_1, \lambda_2$
**Ensure:** Fine-tuned policy $\pi_\theta$ with enhanced representation capabilities
 1: **for** epoch $= 1$ to $N_{\text{epochs}}$ **do**
 2:   **for** batch $\mathcal{B} = \{(q_i, d_i^+, d_i^-)\}_{i=1}^B \sim \mathcal{D}$ **do**
 3:     *// Phase 1: Generate interpretable rationale via policy*
 4:     **for** $i = 1$ to $B$ **in parallel do**
 5:       Generate query rationale: $y_{q_i} =$r$\sim \pi_\theta(\cdot|\mathcal{P}(q_i))$
 6:       Generate negative rationale: $y_{d_i^-} \sim \pi_\theta(\cdot|\mathcal{P}(d_i^-))$
 7:       Sample $K$ positive rationale: $\{y_{d_i^+}^{(k)}\}_{k=1}^K \sim \pi_\theta(\cdot|\mathcal{P}(d_i^+))$
 8:     **end for**
 9:     *// Phase 2: Extract semantic representations via masked pooling*
10:     Encode all rationale: $\mathbf{E} = \pi_\theta(\mathcal{P}(x) \oplus y)$ for all $(x, y)$ pairs
11:     Apply masked mean pooling (Eq. 3):
12:       $\mathbf{h}_{q_i}, \mathbf{h}_{d_i^-}, \{\mathbf{h}_{d_i^+}^{(k)}\}_{k=1}^K \leftarrow \text{MaskedPool}(\mathbf{E})$
13:     *// Phase 3: Compute multi-dimensional contrastive rewards*
14:     **for** $i = 1$ to $B$ **do**
15:       Identify hard negatives: $\mathcal{H}_i = \{j \neq i : \max_l \text{sim}(\mathbf{h}_{q_i}, \mathbf{h}_{d_j^+}^{(l)})\}$
16:       **for** $k = 1$ to $K$ **do**
17:         $\mathcal{R}_{\text{CL}}^{(i,k)} = \text{sim}(\mathbf{h}_{q_i}, \mathbf{h}_{d_i^+}^{(k)}) - \sum_{m=1}^{M_i} \text{sim}(\mathbf{h}_{q_i}, \mathbf{h}_{d_{i,m}^-})$
18:         $\mathcal{R}_{\text{consist}}^{(i,k)} = \frac{1}{K-1} \sum_{j \neq k} \text{sim}(\mathbf{h}_{d_i^+}^{(k)}, \mathbf{h}_{d_i^+}^{(j)})$
19:         $\mathcal{R}_{\text{hard}}^{(i)} = -\frac{1}{B-1} \sum_{\substack{j=1 \\ j \neq i}}^B \max_{1 \leq l \leq K} \text{sim}(\mathbf{h}_{q_i}, \mathbf{h}_{d_j^+}^{(l)})$
20:         $\mathcal{R}_{\text{total}}^{(i,k)} = \mathcal{R}_{\text{CL}}^{(i,k)} + \lambda_1 \mathcal{R}_{\text{consist}}^{(i,k)} + \lambda_2 \mathcal{R}_{\text{hard}}^{(i,k)}$
21:         $\hat{\mathcal{R}}_{\text{total}}^{(i,k)} = \mathcal{R}_{\text{total}}^{(i,k)}/\tau$
22:       **end for**
23:     **end for**
24:     *// Phase 4: Optimize policy via GRPO*
25:     Compute group baseline: $b_i = \frac{1}{K} \sum_{k=1}^K \mathcal{R}_{\text{total}}^{(i,k)}$
26:     Compute advantages: $A^{(i,k)} = \mathcal{R}_{\text{total}}^{(i,k)} - b_i$
27:     Update policy: $\theta \leftarrow \theta + \alpha \nabla_\theta \mathcal{L}_{\text{GRPO}}(\theta)$ where
28:       $\mathcal{L}_{\text{GRPO}} = \sum_{i,k} A^{(i,k)} \log \pi_\theta(y_{d_i^+}^{(k)}|\mathcal{P}(d_i^+))$
29:   **end for**
30: **end for**
31: **return** Optimized policy $\pi_\theta$

---

### A.4.3 CONVERGENCE ANALYSIS

The optimization landscape of our framework benefits from the variance reduction properties of both contrastive learning and policy gradient optimization. The consistency reward acts as a regularizer, bounding the policy updates:

$$\|\nabla_\theta J(\theta)\| \leq C \cdot \mathbb{E}\left[\|\mathcal{R}_{\text{CL}}\| + \lambda_1 \|\mathcal{R}_{\text{consist}}\| + \lambda_2 \|\mathcal{R}_{\text{hard}}\|\right] \tag{27}$$

where $C$ is a constant dependent on the policy parameterization. The consistency term $\lambda_1 \mathcal{R}_{\text{consist}}$ ensures that $\|y_{d_i^+}^{(k)} - y_{d_i^+}^{(j)}\| \leq \epsilon$ for small $\epsilon$, preventing divergent interpretations and ensuring stable convergence.

This theoretical foundation reveals that our approach maintains the discriminative power of contrastive learning while unleashing the generative capabilities of LLMs, enabling the policy to actively interpret and understand text rather than merely discriminate between samples.

## A.5    FURTHER ANALYSIS

### A.5.1    SUBSET OF MTEB BENCHMARK

To enable compute-efficient ablations while preserving coverage across the MTEB taxonomy, we evaluate on a compact, representative subset of 16 datasets spanning seven task families (Table A.5.1): Retrieval (3), Reranking (2), Clustering (3), Pair Classification (1), Classification (3), STS (3), and Summarization (1). We run all ablation studies on this 16-task subset in order to systematically isolate component contributions, probe hyperparameter sensitivity, and compare training strategies under a fixed compute budget, while maintaining fidelity to the full benchmark. The selection balances domain diversity (biomedical, news, open-domain, intent), supervision formats (point-wise, pair-wise, list-wise), and difficulty, yielding stable signals for both representation quality and downstream utility. For each task we report the official metric defined by MTEB (e.g., nDCG@10 for retrieval, MAP for reranking, V-measure for clustering, accuracy/AP for classification and pair classification, and Spearman correlation for STS; summarization uses the SummEval correlation provided by the harness).

| Task | Dataset |
| --- | --- |
| Retrieval (3) | SciFact
ArguAna
NFCorpus |
| Reranking (2) | StackOverflowDupQuestions
SciDocsRR |
| Clustering (3) | BiorxivClusteringS2S
MedrxivClusteringS2S
TwentyNewsgroupsClustering |
| Pair Classification (1) | SprintDuplicateQuestions |
| Classification (3) | Banking77Classification
EmotionClassification
MassiveIntentClassification |
| STS (3) | STS17
SICK-R
STSBenchmark |
| SummEval (1) | SummEval |
| Overall | 16 datasets |

Table 5: Subset of MTEB tasks used for our ablations and analysis.

### A.5.2    EFFICIENCY ANALYSIS

We decompose end-to-end latency into encoding ($T_{\text{encode}}$), generation ($T_{\text{gen}}$), and matching ($T_{\text{match}}$). Figure 6(a) shows that for generative pipelines (Base and GRACE), $T_{\text{gen}}$ overwhelmingly dominates the budget, whereas $T_{\text{encode}}$ and $T_{\text{match}}$ are comparatively small. As a result, encoder-style approaches (BERT and Direct forward method) achieve much lower latency since they avoid generation. Increasing the decoding budget from $G{=}256$ to $G{=}512$ further amplifies $T_{\text{gen}}$ with diminishing returns, indicating a practical knee around $G{\approx}256$.

Figure 6(b) summarizes the quality–latency trade-off. BERT and Direct forward method occupy the ultra–low-latency region but underperform on accuracy, while generative methods deliver higher quality at greater cost. At matched $G$, GRACE consistently shifts the Pareto frontier upward relative to Base, yielding better accuracy without extra latency and making GRACE at $G{=}256$ a strong default for balanced deployments.

Pragmatically, because $T_{\text{gen}}$ dominates end-to-end latency, the most effective lever is to accelerate generation. Deploying on newer accelerators and using optimized inference stacks with fused attention kernels, paged/continuous batching, and CUDA Graphs can materially reduce $T_{\text{gen}}$ at fixed

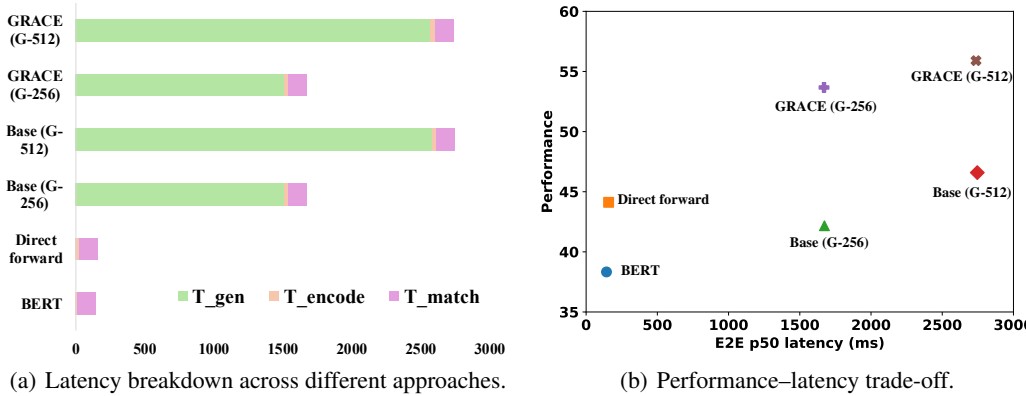

(a) Latency breakdown across different approaches.

(b) Performance–latency trade-off.

Figure 4: Efficiency comparison of different embedding approaches.

$G$. Additional gains come from KV-cache optimizations, speculative/assisted decoding, and low-precision execution (FP8/INT8/INT4) that preserve accuracy under our evaluation budgets. In short, improving generation throughput shifts the entire quality–latency curve downward without changing the training procedure, making GRACE at $G{=}256$ even more attractive for balanced deployments.

### A.6 JOINT ANALYSIS OF GENERATIVE AND EMBEDDING PERFORMANCE

Figure 1 illustrates the positioning of our proposed GRACE models in the joint landscape of generative and embedding performance. Existing encoder-only methods such as E5 Wang et al. (2024), LLM2Vec BehnamGhader et al. (2024), and text-embedding-3 [1] achieve strong embedding quality but exhibit minimal generative competence, while instruction-tuned decoders (e.g., GPT OpenAI (2023), Claude Anthropic (2024), Gemini Gemini Team (2025), Grok xAI (2025), Deepseek DeepSeek-AI et al. (2025)) demonstrate strong generative capabilities but poor embedding performance. This highlights the long-standing trade-off between the two dimensions.

By contrast, the GRACE family consistently shifts base models upward in embedding quality while largely preserving their generative competence. For example, GRACE-1.5B and GRACE-3B improve the embedding strength of Qwen2.5-1.5B and LLaMA-3.2-3B by more than 15 % on average without sacrificing generative ability. Similarly, GRACE-4B substantially enhances Qwen-3-4B, pushing it into a previously unattained regime of balanced performance. These results underscore the effectiveness of our generative-contrastive optimization framework in bridging the gap between high-quality embeddings and generative reasoning.

#### A.6.1 TRAINING PROGRESSION ANALYSIS

As shown in Figure 5, both task performance and response length increase steadily with more training steps. The accuracy on subtasks follows a consistent upward trajectory, reflecting that extended training directly strengthens the models' ability for representation. At the same time, the generated responses become progressively longer. Importantly, this lengthening indicates that the models are producing answers with richer information density and more explicit reasoning chains. In earlier stages, responses tend to be short and often incomplete, whereas later stages exhibit structured explanations that combine factual correctness with reasoning depth.

#### A.6.2 EFFECTS OF VARIOUS REPRESENTATION APPROACHES

We investigate the impact of different token representation methods on model performance across both supervised and unsupervised settings. Figure 6 presents a comparative analysis of four representation approaches: EOS token, Max Pooling, Mean Pooling from the last layer (LL), and Mean Pooling from the penultimate layer (PL). The results demonstrate that mean pooling approaches consistently outperform both EOS token and max pooling methods across all model variants in both settings. Mean

[1]https://openai.com/index/new-embedding-models-and-api-updates/

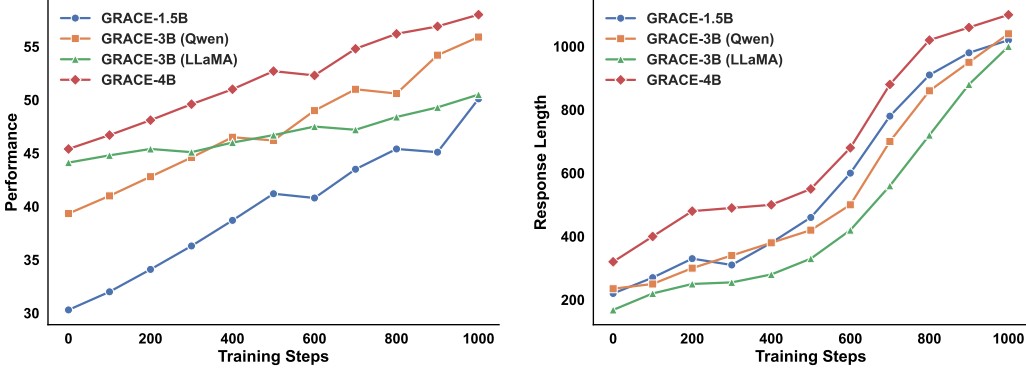

(a) Accuracy progression across training steps.

(b) Response length progression across training steps.

Figure 5: Training progression of GRACE models. Left: accuracy on subtasks steadily improves with more training steps. Right: response length also increases, reflecting enhanced information density and richer reasoning chains.

pooling from the last layer and penultimate layer achieve remarkably similar performance levels and slightly higher in unsupervised models, while EOS token and max pooling show substantially lower performance. The minimal performance difference between LL and PL mean pooling suggests that both layers capture equally effective representations, which aligns with Skean et al. (2025), indicating that comprehensive token aggregation through mean pooling is crucial for optimal performance.

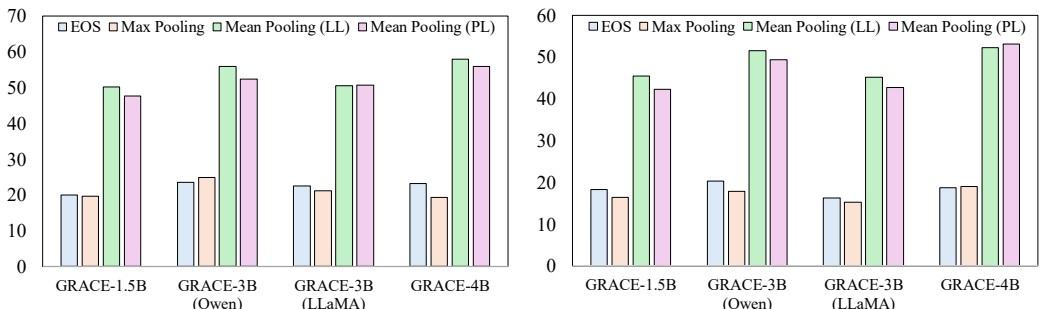

(a) Performance of various representation approaches in supervised fine-tuning.

(b) Performance of various representation approaches in unsupervised training.

Figure 6: Comparison of different token representation methods across GRACE model variants. Mean pooling from both last layer (LL) and penultimate layer (PL) consistently outperform EOS token and max pooling approaches in both supervised and unsupervised settings.

### A.6.3 ADDITIONAL TRAINING DETAILS

We incorporate length regularization to prevent degenerate solutions where the model generates excessively long responses without meaningful content:

$$
\mathcal{R}_{\text{final}}^{(i,k)} = \begin{cases} \hat{\mathcal{R}}_{\text{total}}^{(i,k)} & \text{if } |y_{d_i^+}^{(k)}| < L_{\max} \text{ or } y_{d_i^+}^{(k)}[-1] = \text{EOS} \\ -\gamma & \text{otherwise} \end{cases} \tag{28}
$$

where $L_{\max}$ is the maximum allowed sequence length and $\gamma$ is a penalty coefficient for over-length generations.

### A.6.4 ADDITIONAL IMPLEMENTATION DETAILS

Training uses the AdamW Loshchilov & Hutter (2019) optimizer with a fixed learning rate of $1 \times 10^{-6}$ and no warmup schedule. The maximum prompt length is set to 1024 tokens and the maximum

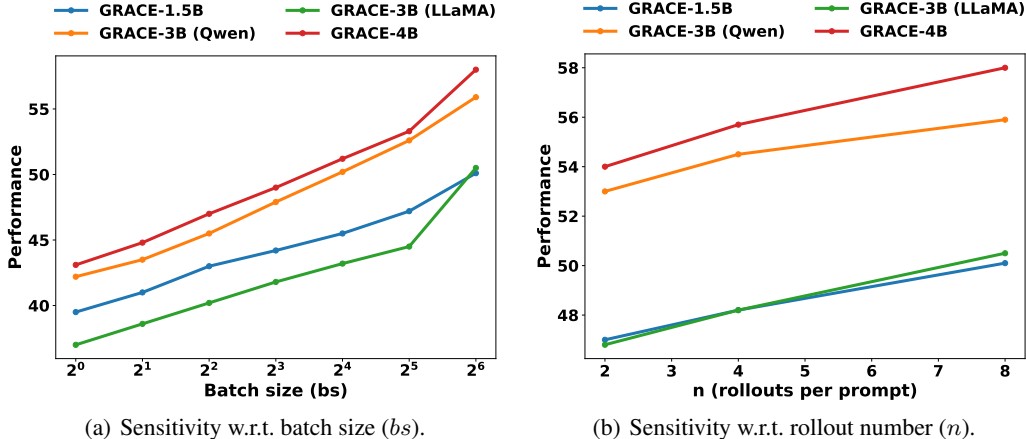

(a) Sensitivity w.r.t. batch size ($bs$).   (b) Sensitivity w.r.t. rollout number ($n$).

Figure 7: Hyperparameter sensitivity for GRACE models. Both curves show monotonic performance improvements. Left: performance increases monotonically with larger $bs$. Right: higher $n$ yields steady gains with reduced variance at larger values. The best setting is $bs = 64$ and $n = 8$, with further improvements possible given more GPU resources.

response length to 2048 tokens, with over-length sequences truncated on the right side. A length penalty $\gamma = 1.0$ is applied when responses reach the maximum length without emitting an EOS token. For GRPO, we generate $K = 8$ rollouts per positive document using temperature sampling with $T = 1.0$, and compute advantages relative to a mean baseline without normalization. The consistency weight $\lambda_1 = 0.2$, and the hard negative weight $\lambda_2 = 0.2$. The scaling $\tau$ for our training is set to 10. Multi-GPU training is managed with Fully Sharded Data Parallel (FSDP) Zhao et al. (2023), with parameter and optimizer state sharding to alleviate memory constraints. Training-time generation leverages vLLM [2] with tensor-parallel size 2 and 50% GPU memory utilization in eager mode for stability. Instruction tokens are automatically detected and excluded from pooling operations to prevent contamination in representation extraction. At inference time, we adopt the same generation and pooling configurations as training to ensure consistency.

### A.6.5 HYPERPARAMETER SENSITIVITY ANALYSIS

We analyze the sensitivity of model performance to two training-time hyperparameters: the batch size $bs$ and the number of rollouts per prompt $n$. Overall, all models exhibit a monotonic improvement with increasing batch size. Figure 7 (right) varies $n \in \{2, 4, 8\}$ and shows consistent, monotonic gains across models, with diminishing variance at higher $n$. In practice, we find $bs \geq 32$ and $n \geq 4$ to be a good Pareto choice under typical compute budgets, while larger settings such as $bs = 64$ and $n = 8$ deliver the best observed performance. Further improvements are likely possible given more GPU resources.

### A.6.6 CASE STUDY

**Output Evolution During Training** Table 6 illustrates how outputs evolve with training. The base model provides only a brief topical summary, while step 200 introduces a structured listing with concrete figures. By step 600, the output grows substantially longer and begins to integrate historical context, crises, and institutional concepts. At step 1000, the response is the most comprehensive: it connects anecdotes with broader political and constitutional developments, offering a coherent narrative. This progression shows that training not only increases output length but also enhances contextualization and information density, reflecting richer internal representations.

**In-depth Analysis of Model Response Patterns** We highlight two representative cases that demonstrate the model's response patterns across different input types, shown in Table 7. For

---

[2]https://github.com/vllm-project/vllm

| **Original Doc:** BBC - History - British History in depth: Prime Ministers and Politics Timeline On This Day Prime Ministers and Politics Timeline Do you know which prime minister brought 'fallen women' to 10 Downing Street? Or which one fought a duel? Or who was known as 'the Goat'? Take a political journey through nearly 300 years of high ideals and low cunning, from Gordon Brown to the first man to hold prime minister... |
|---|

| **Base Model (480 tokens)** | **Training Step 200 (562 tokens)** | **Training Step 600 (824 tokens)** | **Training Step 1000 (1051 tokens)** |
|---|---|---|---|
| "The text is comprised of an interactive timeline on the history of **British Prime Ministers**, a concise biography of several past **Prime Ministers**, and a broader overview of the office." ... "It begins with timed anecdotes highlighting key figures and events from **1681** to the present day." | "Provides an overview of several key **British prime ministers** and the history of the office." ... "Starts with **Margaret Thatcher**, Britain's first **female PM**, and also mentions **Robert Walpole** and **Henry Campbell-Bannerman**." ... "Walpole is seen as the first PM, Campbell-Bannerman the first formally called so." | "Informative passage about key British Prime Ministers, such as **Margaret Thatcher**, **James Callaghan**, and **Harold Wilson**." ... "Traces the evolution of the role from **18th century cabinet system**, with Walpole as early figure." ... "Highlights crises like Harold Wilson's **industrial disputes** and **economic instability**." ... "Explains **parliamentary accountability** and **collective responsibility**." | "Provides a comprehensive understanding of the office of the **Prime Minister** and its historical context." ... "Roots traced back to **Robert Walpole**, later evolution shaped by constitutional changes." ... "Mentions modern figures like **Thatcher** and **Tony Blair**, situating them in **UK's political-economic**." ... "Discusses powers and constraints of **PM**, from cabinet appointments to parliamentary limits." ... "References anecdotes (e.g., '**fallen women**') as outdated, while contextualizing gender role shifts." |

Table 6: Case study for output evolution. As training progresses, outputs grow from brief topic recognition to structured lists, then detailed coverage, and finally full contextualization.

| **Doc Case (Belt and Road Initiative)** | **Query Case (Machu Picchu)** |
|---|---|
| **Original Document:** "policy coordination is an important guarantee for implementing the Initiative... We should carry forward the spirit of friendly cooperation of the Silk Road by promoting cultural and academic ..." | **Original Query:** "In which country would you find the ruins of the ancient Inca city known as Machu Picchu?" |
| **Model Response:** "The Belt and Road Initiative is an extensive cooperation project proposed by China... *Policy Coordination* involves inter-governmental cooperation, expansion of shared interests, and enhancing mutual trust... *People-to-People Bonds* promote cultural exchange, volunteer services, and media cooperation... The initiative faces challenges such as debt sustainability and geopolitical risks..." | **Model Response:** "Machu Picchu is located in **Peru**... situated in the Andes Mountains, Cusco Region, at about 2,430m elevation... rediscovered in 1911 by Hiram Bingham... a UNESCO World Heritage Site and major tourist destination..." |

Table 7: Representative cases of model responses: left column shows document understanding (doc case), right column shows query expansion (query case).

a long document on the "Belt and Road Initiative", the model goes beyond paraphrase to produce a compact outline (policy coordination, infrastructure and finance, people-to-people links, geopolitical constraints), turning an abstract preface into a structured analysis. For a minimal factual query (e.g., "Where is Machu Picchu?"), it answers precisely while adding concise, high-signal context (Peru; Andean setting; Inca and UNESCO notes; basic access considerations), avoiding unnecessary narrative. This adaptive behavior yields information-dense, well-factored texts whose embeddings align with latent topics and relations, improving separability for retrieval, clustering, and pair classification.

