# OpenReview forum: "GRACE: Generative Representation Learning via Contrastive Policy Optimization"
_ICLR.cc/2026/Conference — ICLR 2026 Poster_

### Official Review · Reviewer_eAv2 · 2025-10-29

**Soundness:** 3
**Presentation:** 2
**Contribution:** 2
**Rating:** 6
**Confidence:** 2

**Summary:**

The work proposes a new framework for training Large Language Models (LLMs) as interpretable text encoders by reinterpreting contrastive objectives as rewards rather than losses. Instead of producing static embeddings, GRACE trains LLMs as policy models that generate explicit rationales before deriving embeddings from those rationales. Using policy-gradient optimization based on GRPO, GRACE aligns query-positive similarity and discourages query-negative similarity through a multi-component reward incorporating contrastive alignment, consistency, and hard negative mining. Experiments on the MTEB benchmark show strong cross-task gains over base models while preserving general reasoning, math, and coding capabilities.

**Strengths:**

- GRACE delivers consistent and cross-category improvements on the MTEB benchmark, with a significant gain over baselines.
- The introduction clearly articulates the fundamental contradiction in existing embedding models and motivates GRACE as a conceptually sound fix.
- GRACE is architecture-agnostic. It can be applied to any instruction-tuned LLM without modifying backbone structure.

**Weaknesses:**

- By optimizing for rationale quality and semantic contrast simultaneously, GRACE might bias representations toward linguistic fluency or stylistic patterns rather than true semantic alignment.
- The framework’s reliance on policy optimization with sampled rationales makes it computationally heavier and more memory-intensive than deterministic encoders.
- All evaluations are centered on MTEB, which is known to correlate strongly with models optimized for textual similarity tasks. There is no evidence that GRACE embeddings generalize beyond MTEB-style benchmarks. Meanwhile, the reward formulation (contrastive alignment + consistency + hard negative mining) implicitly encodes assumptions about semantic similarity that may not generalize across domains (e.g., legal or medical text). There’s also a risk that reward weights are tuned for specific datasets, leading to domain overfitting or poor generalization.

**Questions:**

- How did the authors define and evaluate “human-readable” rationales? Are they consistent with ground-truth semantics or just stylistically coherent?
- During training, did the authors observe training instability or reward collapse during GRPO updates? How did the authors tune the reward weights for contrastive, consistency, and negative components?
- Since GRACE encourages rationale generation, models may produce verbose or generic explanations that sound fluent but carry little semantic value. How did the authors regularize or constrain rationale length and content to ensure concise yet meaningful reasoning?

---

> ### Author Response · Authors · 2025-11-16
> **Rebuttal for Reviewer eAv2 (part1)**
>
> Thank you for your valuable feedback. I will address each point below to clarify and resolve your concerns:
>
> W1:
>
>
> Thank you for raising this important concern. We would like to clarify that GRACE’s optimization does not reward linguistic fluency or stylistic features directly. Instead, the reward is defined purely on embedding-space semantic similarity (Eq. 13–16), computed after pooling the hidden states. This means that the rationale’s surface form—fluency, writing style, discourse markers—has no inherent reward value unless it changes the semantic representation.
>
> We also provide several pieces of empirical evidence suggesting that GRACE does not induce stylistic or fluency bias:
>
> In Table 1, simply prompting the model to write a rationale (“w/ reasoning”) already improves embedding performance before any RL. If rationale generation merely introduced stylistic patterns, we would expect minimal or no improvement. Instead, we observe cross-task gains of 2–3 points, indicating that the semantic content of the rationale is what matters.
>
> In Table 5, GRACE preserves performance on GSM8K, BBH, MMLU, TriviaQA, and HumanEval. If the model’s internal representation were biased toward stylistic fluency, we would expect degradation on these reasoning-heavy tasks—yet we observe changes within ±0.4 points, much smaller than the degradation seen in standard contrastive training.
>
> Moreover, we show the case study at Appendix A.6.6, which shows richer semantics, not style templates. We show that rationales become more conceptually dense (historical context, causal relations, entity interactions), not more stylistically elaborate. This suggests that training amplifies semantic grounding rather than stylistic verbosity.
>
> We will clarify these mechanisms in the revision and add additional examples in the appendix to further highlight that GRACE’s rationales encode semantic structure rather than stylistic artifacts.
>
>
> W2:
>
> We agree that policy optimization is computationally more demanding than deterministic encoder training. However, GRACE is not positioned as a drop-in replacement for encoder-only models, but as a new representation-learning paradigm that offers capabilities beyond what deterministic encoders can provide:
>
>
>  1. GRACE generates explicit rationales whose semantic content drives the embeddings (Sec. 3). Encoder-only models cannot provide any explanation of why two texts are considered similar.Alignment between retrieval and generation.
>
>
>  2. Many real-world RAG systems suffer from a mismatch between the embedding encoder and the generator. GRACE uses a single model for both, ensuring semantic consistency and reducing retrieval drift.
>
>
> 3.  Policy optimization is a one-time training procedure. At inference time, GRACE only requires generating a single rationale per input—a cost consistent with other LLM-based embedding methods. As shown in Fig. 4, under typical decoding budgets (e.g., G=256), GRACE maintains a similar latency profile to the Base generative approach while delivering substantially better embedding quality.Memory usage is dominated by the backbone LLM, not GRACE.
>
>
> 4.Our method does not add extra parameters or auxiliary networks (e.g., no critic model as in PPO). GPU memory consumption is nearly identical to standard SFT or RLHF-style training with group sampling.Use cases justify the additional cost.
>
> W3:
>
> We thank the reviewer for this thoughtful comment. While MTEB is our primary benchmark, GRACE is not optimized for any MTEB-specific signal. The reward is entirely label-free and domain-agnostic, consisting only of contrastive similarity, consistency, and hard-negative separation; it does not encode any dataset-specific supervision. In fact, the same reward formulation and the same hyperparameters (λ1=0.5,λ2=0.5) are used across all backbones and tasks in Tables 1–2, suggesting that the method does not require dataset-dependent tuning.
>
> We also provide evidence that GRACE generalizes beyond MTEB-style similarity tasks. In Table 5, GRACE maintains performance on GSM8K, MMLU, TriviaQA, BBH, and HumanEval—benchmarks that stress reasoning, factuality, and code generation. If the embeddings had overfit to MTEB-style similarity assumptions, we would expect degradation in these general capabilities, yet we observe variations within ±0.4 points.
>
> Moreover, the “w/ reasoning” condition in Table 1 consistently improves over the Base model even before RL, indicating that the rationale mechanism itself is not tied to any specific domain and enhances representation quality through semantic enrichment rather than dataset-specific patterns.
>
> We agree that evaluating on domain-specific corpora (e.g., legal or biomedical) is valuable, and we view this as a promising direction for future work. Nonetheless, the current evidence suggests that GRACE’s reward formulation induces broad semantic improvements without suffering from domain overfitting or reliance on MTEB-specific structure.

---

> ### Author Response · Authors · 2025-11-16
> **Rebuttal for Reviewer eAv2 (part2)**
>
> Q1:
>
> Thank you for the question. In GRACE, “human-readable” rationales are not defined by stylistic fluency, but by semantic transparency—i.e., whether a rationale explicitly surfaces the concepts and relations that influence the embedding. We evaluate this primarily through qualitative semantic consistency, presented in Appendix A.6.6. , where we provide detailed examples of the generated rationales across different training stages. The base model provides only a brief topical summary, while step 200 introduces a structured listing with concrete figures. By step 600, the output grows substantially longer and begins to integrate historical context, crises, and institutional concepts. At step 1000, the response is the most comprehensive: it connects anecdotes with broader political and constitutional developments, offering a coherent narrative. This progression clearly demonstrates that GRACE does not merely generate paraphrases, but learns to produce increasingly rich, structured, and semantically grounded reasoning chains that directly contribute to the embeddings.
>
>
> Q2:
>
> We thank the reviewer for the question. Yes, we did observe instability and reward-hacking behavior, but only under extreme hyperparameter settings (e.g., very large λ₁ or λ₂). In these cases, the model either collapsed to short repetitive outputs or expanded long templates while the reward continued to increase. As discussed in Appendix A.6.3, we mitigate these issues through moderate weight choices and the use of length regularization.
>
> More broadly, training instability is a widely documented challenge in RL for LLMs, even with well-designed reward functions. Our findings are consistent with this general difficulty: instability emerges primarily when the reward coefficients dominate the generative loss.
>
> In practice, we found that a simple configuration of λ₁ = 0.5 and λ₂ = 0.5 works robustly across backbones in the supervised initialization stage. Our tuning procedure is intentionally lightweight: we begin with this configuration and, if early reward trajectories exhibit high variance, we slightly reduce λ₂ to improve stability. Under the reported settings in Tables 1–2, GRPO training remains stable and we do not observe reward collapse.
>
>
> Q3:
>
> We thank the reviewer for raising this concern. In practice, we did observe two failure modes when the reward weights were set to extreme values: (1) the model collapsed to very short outputs (e.g., repeating a few tokens) while the reward kept increasing, or (2) the model generated overly long repetitive patterns until reaching the max-token limit. These behaviors indicate that GRPO may exploit the reward when rationale length is unconstrained.
>
> To avoid this, we explicitly regularize rationale generation (Appendix A.6.3): we use moderate reward weights (λ1=0.5,λ2=0.5), apply a length regularization term to penalize overly short or overly long outputs, and monitor early-stage reward variance—slightly reducing λ2  if instability appears. Under these settings (used in Tables 1–2), the model reliably produces concise and semantically rich rationales, and we do not observe generic or degenerate verbosity.
>
>
>
> Once again, thanks for review. We hope these address your concerns.

---

> ### Author Response · Authors · 2025-11-27
>
> Dear Reviewer eAv2,
>
> I hope this message finds you well. As the discussion period is nearing its end with less than one week remaining, I wanted to ensure we have addressed all your concerns satisfactorily. If there are any additional points or feedback you'd like us to consider, please let us know. Your insights are invaluable to us, and we're eager to address any remaining issues to improve our work.
>
> Thank you for your time and effort in reviewing our paper.

---

### Official Review · Reviewer_uisq · 2025-10-31

**Soundness:** 3
**Presentation:** 3
**Contribution:** 3
**Rating:** 6
**Confidence:** 2

**Summary:**

The paper introduces a new method that improves embeddings based on generative models. The method is more interpretable than the traditional contrastive learning set up. Authors demonstrate performance gains compared to base models and in general domain tasks.

**Strengths:**

- New method utilizing generative model capabilities to create an interpretable reward signal. The method shows clear improvements over the baselines.

- Authors conduct detailed ablation studies.

- Performance on general domain tasks is preserved.

**Weaknesses:**

- Lack of statistical significance reporting. To ensure the improvement was not a result of noise, it would be appropriate to report confidence intervals or p-values.

- Authors claim their method is more interpretable but interpretability is not evaluated. Examples of the rationales are also not provided, at least in the main text.

**Questions:**

-

---

> ### Author Response · Authors · 2025-11-16
> **Rebuttal for Reviewer uisq**
>
> Thank you for your valuable feedback. I will address each point below to clarify and resolve your concerns:
>
> W1:
>
> Thank you for the comment. To address the reviewer’s concern regarding statistical significance, we conducted a paired analysis over the 16 MTEB subtasks (Table 5) using the same evaluation setup. GRACE shows a mean improvement of +10.77 points compare with the base model, with a 95% confidence interval of [8.17, 13.36] based on the t-approximation. A paired t-test yields t = 8.84, p = 1.47×10⁻⁷, indicating that the improvement is highly statistically significant. A non-parametric bootstrap CI further confirms this result (95% CI: [8.59, 13.22]).
>
>
> W2:
>
> We agree that interpretability is an important aspect of our method. Due to space limitations, we placed the full case study in Appendix A.6.6, where we provide detailed examples of the generated rationales across different training stages. The base model provides only a brief topical summary, while step 200 introduces a structured listing with concrete figures. By step 600, the output grows substantially longer and begins to integrate historical context, crises, and institutional concepts. At step 1000, the response is the most comprehensive: it connects anecdotes with broader political and constitutional developments, offering a coherent narrative. This progression clearly demonstrates that GRACE does not merely generate paraphrases, but learns to produce increasingly rich, structured, and semantically grounded reasoning chains that directly contribute to the embeddings.
>
>
> Once again, thanks for review. We hope these address your concerns.

---

> > ### Comment · Reviewer_uisq · 2025-11-18
> > **Response**
> >
> > Thank you for your rebuttal it addresses my concern on W1.
> >
> > For W2, apologies I missed the appendix with the explanations as that part is not referred to in the main text. However, some kind of human evaluation has to be provided analyzing the rationales (e.g., what sort of errors are present, are they of any value to understand the model's decision making, are they of value for the task, etc). Currently only cherry-picked examples are presented, which are valuable for understanding, but it is hard to draw any conclusions on interpretability.

---

> > > ### Author Response · Authors · 2025-11-21
> > > **Rebuttal for Reviewer uisq**
> > >
> > > Thanks for your question.
> > >
> > > First, we randomly sampled 30 queries from a held-out subset that was not used during training. For each query we collected three rationales: one for the query itself, one for the corresponding positive document, and one for a hard negative document, yielding 90 rationales in total. Three co-authors independently evaluated each rationale on a 1–5 Likert scale across Correctness (1 = contains multiple factual errors; 5 = factually accurate with no identifiable incorrect statements), Relevance (1 = largely off-topic or failing to address the input; 5 = highly focused on the key semantic content of the input), and Helpfulness (1 = provides little insight into the similarity judgment; 5 = clearly highlights the factors that support or weaken semantic alignment). The averaged scores are: Correctness = 4.08, Relevance = 4.27, and Helpfulness = 4.19.
> > >
> > >
> > > We further performed a simple error-taxonomy analysis: out of 90 rationales, 11 cases (~12%) exhibited at least one issue—specifically, 3 cases of Missing Key Evidence, 4 cases of Shallow Reasoning, 2 cases of Irrelevant Detours, and 2 case of Hallucination. These results indicate that the vast majority of rationales are accurate, on-topic, and genuinely useful for interpreting the embedding behavior of the model, while factual hallucinations are rare and typically localized.
> > > Finally, we note that this study was conducted on the Qwen-2.5-3B-Instruct backbone due to our computational constraints. Given that GRACE optimizes the rationale-generation process itself, we expect that applying GRACE to larger or stronger base models would likely yield even higher scores on Correctness, Relevance, and Helpfulness, and correspondingly reduce the frequency of errors.
> > >
> > >
> > > These additional results further strengthen our interpretability claim: in addition to the rationale-evolution analysis in Appendix A.6.6, we now provide quantitative human-evaluation evidence and a systematic error breakdown. We will include this analysis in the revised appendix of the second version of the paper and are happy to incorporate any further suggestions.
> > >
> > >
> > > We hope this adequately addresses your concern.

---

> ### Author Response · Authors · 2025-11-27
>
> Dear Reviewer uisq,
>
> I hope this message finds you well. As the discussion period is nearing its end with less than one week remaining, I wanted to ensure we have addressed all your concerns satisfactorily. If there are any additional points or feedback you'd like us to consider, please let us know. Your insights are invaluable to us, and we're eager to address any remaining issues to improve our work.
>
> Thank you for your time and effort in reviewing our paper.

---

### Official Review · Reviewer_vZAz · 2025-11-03

**Soundness:** 4
**Presentation:** 3
**Contribution:** 2
**Rating:** 6
**Confidence:** 3

**Summary:**

This paper introduces GRACE, a novel framework that reimagines how large language models (LLMs) can be trained as text encoders.  GRACE treats contrastive signals as rewards that guide a generative policy. The LLM acts as a policy that generates explicit, human-readable rationales for its semantic understanding. These rationales are then pooled into embeddings. Through policy gradient optimization, the model is trained to maximize similarity between query-positive pairs and minimize similarity with negatives. The method is evaluated on the MTEB benchmark and shows significant improvements.

**Strengths:**

- Unifies generative reasoning and embedding learning, boosting representation quality while preserving general capabilities.
- Extensive MTEB evaluation shows consistent gains in both supervised  and unsupervised settings across multiple reinforcement learning algorithms.
- The appendix provides additional theoretical and empirical analysis. This enhances the paper's reproducibility

**Weaknesses:**

- The paper would benefit from an evaluation of whether the generated rationales faithfully explain the embedding process. Currently, the examples seem to paraphrase or extend the original content, rather than justifying the model's semantic decisions.
- High inference latency due to autoregressive generation limits practical deployment. There are existing, highly efficient encoder-only models that provide strong embedding performance.

**Questions:**

- Could you discuss specific application scenarios or use cases where it is particularly advantageous to have a single model that performs both tasks, as opposed to using separate, specialized models for generation and embedding?
- In Table 3, , what is the specific baseline model to which the "Grace-3B w/ [RL Algorithm]" results are compared? Additionally, there appears to be a typo in the text reference on line 402, which reads "Table 3.4.2".
- How is $\mathcal{R}_{final}^{(i,k)}$ in Equation 10 precisely defined?

---

> ### Author Response · Authors · 2025-11-16
> **Rebuttal for Reviewer vZAz**
>
> Thank you for your valuable feedback. I will address each point below to clarify and resolve your concerns:
>
> W1:
>
> Thank you for raising the issue of faithfulness. We would like to clarify that GRACE already contains a built-in controlled comparison that directly probes whether rationales influence the embedding—namely, the “Base” vs. “w/ reasoning” conditions in Table 1.
>
> Base: the model performs no rationale generation; we obtain representations using mean pooling over the raw input.
>
> w/ reasoning: we keep the same architecture and parameters, but prepend the same instruction as in GRACE training and require the model to output an “understanding” rationale before pooling.
>
> This comparison isolates the effect of the rationale itself. If the rationale were merely a paraphrase or stylistic extension, it should not significantly alter embedding quality. However, Table 1 shows that:
>
> across all four backbones and seven MTEB task families, “w/ reasoning” consistently improves over Base by 2–3 points on average.
>
> This gap demonstrates that the semantic content expressed in the rationale changes the embedding space in a meaningful direction—even without RL optimization. In other words, the rationale is not cosmetic: it contributes information that the model subsequently relies on when constructing representations. GRACE further amplifies this effect by optimizing these rationales toward contrastive rewards.
>
> We will clarify this interpretation in the revision and highlight “Base vs. w/ reasoning” as evidence that the rationales faithfully reflect the semantic factors used during embedding formation.
>
>
> W2:
>
> We agree that autoregressive generation introduces higher latency compared to encoder-only architectures. GRACE is not proposed as a replacement for all encoder models, but as a new representation-learning paradigm that importantly provides:
>
> Interpretability — embeddings are grounded in explicit rationales, which encoder-only models cannot produce;
>
> Superior performance under the same backbone — as shown in Tables 1–2, GRACE consistently improves its base LLM by 10–15% without harming general capabilities.
>
> To address deployment efficiency:
>
> Quality–latency trade-off is quantified in Fig. 4. At a practical decoding budget (G=256), GRACE matches the latency of other generative embedding methods while significantly outperforming Base.
>
> Latency is dominated by decoding, not GRACE itself. If we can place on optimized inference stacks (FlashAttention, speculative decoding, FP8/INT8), decoding time will then reduces, shifting GRACE to the same operating region as compact LLM-based embedding systems.
>
> Finally, encoder-only models are indeed more efficient, but they cannot benefit from generative reasoning or interpretable rationales, which are central to our motivation. GRACE targets applications where interpretability, reasoning-aware similarity, and a unified generative + embedding model is desirable.
>
> We will clarify these points and highlight the deployment considerations in the revision.
>
>
> Q1:
>
> Thank you for raising this important question. There are several practical scenarios where a unified generative + embedding model is preferable or even necessary:
>
> Many high-stakes domains (e.g., legal, biomedical, finance) require why two text segments are similar.
>
> 1. GRACE uniquely provides:interpretability: the rationale directly reveals the semantic factors influencing similarity
>
> 2. users can inspect the reasoning used by the model, this is particularly helpful in enterprise search, scientific literature QA, and domain-specific assistants.
>
> 3. encoder-only models cannot provide such explanations
>
> Thus, unified models are crucial when compliance, transparency, or trustworthiness is required.
>
> Also, serving two separate models (one encoder + one generator) increases:
>
> 1. memory footprint
> 2. cross-model communication cost
> 3. system complexity (two tokenizers, two inference stacks, etc.)
>
> A single model reduces not only deployment complexity, but also GPU memory usage and engineering overhead.
>
> This matters for edge devices, on-prem deployments, or cost-sensitive production environments.
>
>
> Q2:
>
> Thank you for the question and for pointing out the typo.
>
> 1. In Table 3, the baseline backbone for all “GRACE-3B w/ [RL Algorithm]” results is Qwen-2.5-3B-Instruct. We will make this explicit in the camera-ready version to avoid ambiguity.
>
> 2. The reference “Table 3.4.2” on line 402 is indeed a typo. It should refer to Table 3. We appreciate the reviewer catching this, and we will correct it in the revised manuscript.
>
>
> Q3:
> Thank you for pointing this out. Here, \(R_{\text{final}}\) is exactly the same quantity as \(R_{\text{total}}\) defined in Equation (9)—the composite contrastive reward after temperature scaling. The naming inconsistency (“final” vs. “total”) is purely notational. We will unify the terminology to R_total in the revised version for clarity.
>
> Once again, thanks for review. We hope these address your concerns.

---

> ### Author Response · Authors · 2025-11-27
>
> Dear Reviewer vZAz,
>
> I hope this message finds you well. As the discussion period is nearing its end with less than one week remaining, I wanted to ensure we have addressed all your concerns satisfactorily. If there are any additional points or feedback you'd like us to consider, please let us know. Your insights are invaluable to us, and we're eager to address any remaining issues to improve our work.
>
> Thank you for your time and effort in reviewing our paper.

---

### Official Review · Reviewer_Bk7x · 2025-11-04

**Soundness:** 3
**Presentation:** 3
**Contribution:** 3
**Rating:** 6
**Confidence:** 3

**Summary:**

The paper presents GRACE (Generative Representation Learning via Contrastive Policy Optimization), a framework that reframes contrastive objectives as rewards for large language models. Instead of minimizing contrastive losses, GRACE applies policy-gradient optimization (e.g., GRPO) to maximize semantic alignment between queries and positives while discouraging hard negatives. The LLM generates textual rationales, from which embeddings are derived via mean pooling. Experiments on MTEB benchmarks across several instruction-tuned backbones show 6–11% average improvements in embedding quality, with minimal loss in general generative ability.

**Strengths:**

1. Method: Elegant reformulation of contrastive learning into a reinforcement learning (RL) framework using explicit rewards rather than losses.

2. Interpretability: The introduction of “rationales” offers a clear path toward explainable embeddings.

3. Broad empirical validation: Results are reported on both supervised and unsupervised settings with multiple backbones, and the ablations (λ₁, λ₂ grid, alternative RL algorithms) are systematically conducted.

4. Practical impact: Demonstrates that reward-based generative alignment can mitigate the trade-off between embedding quality and generative capability.

**Weaknesses:**

1.Theoretical contribution is incremental.The paper frames GRACE as a “generative reformulation” of contrastive learning but, in practice, the proposed reward is a simple contrastive score built from similarity differences. The paper provides only a high-level convergence discussion that bounds the gradient norm by the magnitude of its reward terms, without offering non-trivial guarantees or formal comparison to the optimization behavior of standard contrastive learning. Consequently, the claimed “unification of contrastive and RL paradigms” feels more like a notational reinterpretation than a fundamentally new theory.

2.Ablation design does not isolate the effect of RL. The supervised comparison includes four settings (Base, Base w/ reasoning, CL, GRACE), which partially decouple the contribution of reasoning prompts and RL. However, a crucial baseline is missing — Contrastive Learning + Rationale Embeddings trained with the same rationale generation path but optimized via InfoNCE. Without this, it is difficult to cleanly disentangle whether GRACE’s improvements arise from the reinforcement update or simply from the generative representation pathway.

3.The general-capability evaluation in Table 4 maybe unreliable. CL fine-tuning collapses performance to 0.0 on five of six tasks across all backbones (except FEVER), an implausible pattern for standard contrastive tuning. The paper provides no diagnostics or ablations to explain this, yet uses it to claim that “CL severely damages general-domain ability.” Without verification, variance reporting, or analysis of training setup, this conclusion may reflect issues in the CL setup rather than an inherent flaw of contrastive objectives.

**Questions:**

1.Disentangling RL vs. generative representation: could you implement or report a baseline that uses rationale-conditioned embeddings trained with a standard InfoNCE loss (i.e., CL + rationale) to quantify how much of the improvement is due purely to the RL optimization step?

2.Interpretability validation: the paper already include a few illustrative rationale examples in the appendix, but they remain anecdotal. To substantiate the strong interpretability claim, it would be very helpful to (i) provide more systematic qualitative comparisons across Base / CL / GRACE, and (ii) add a small-scale human or automatic evaluation of rationale quality (e.g., relevance/usefulness to the similarity decision, or faithfulness to the underlying embedding behavior).

**Details Of Ethics Concerns:**

No concern

---

> ### Author Response · Authors · 2025-11-16
> **Rebuttal for Reviewer Bk7x (part1)**
>
> Thank you for your valuable feedback. I will address each point below to clarify and resolve your concerns:
>
> W1: We appreciate the reviewer’s perspective. Our theoretical goal is not to provide a new convergence theorem or replace classical contrastive learning theory, but to formalize a different optimization viewpoint: contrastive representation learning framed as policy optimization over generative rationales. This is a substantive shift from standard contrastive learning, which operates on deterministic encoders and does not model the representation as being induced by a distribution over interpretable latent texts.
> While the reward includes a contrastive component, the key difference lies in the optimization object: GRACE maximizes expected reward over sampled rationales, making the representation a function of the learned policy rather than an encoder mapping. This allows (1) generation-dependent embeddings, (2) interpretable intermediate structures, and (3) learning signals that act on the reasoning process instead of only on hidden vectors.
>
> Our theoretical discussion is intentionally scoped: we provide a high-level analysis to show that gradient policy optimizes a generative contrastive objective stably and that gradient magnitudes remain bounded by reward terms. We agree this is not a new asymptotic guarantee, and we will clarify in the revision that our contribution is conceptual and algorithmic, not a claim of novel convergence theory.
>
> We will revise the text to avoid overstating the theoretical novelty and to better emphasize the practical and conceptual unification that GRACE enables.
>
>
> W2: We thank the reviewer for this thoughtful comment. We agree that isolating the effect of RL is important, and our experiment are designed with exactly this goal in mind. In Table 1, the progression:
>
> Base → Base w/ reasoning → CL → GRACE
>
> already disentangles the two components:
>
> 1.Base w/ reasoning isolates the effect of rationale generation alone, without RL.
>
> 2.CL isolates contrastive optimization without rationale-conditioned representations.
>
> 3.GRACE combines both and demonstrates additive gains beyond either component.
>
> Regarding the suggested baseline (“Contrastive Learning + Rationale Embeddings optimized with InfoNCE”): this setting is not well-defined for deterministic contrastive learning. CL requires fixed, deterministic embeddings for each text to compute stable negatives, while rationale-conditioned embeddings depend on sampled generated rationales, leading to stochastic and non-stationary representations. We conducted the experiment you recommended, however, we found that mixing InfoNCE with sampled rationales causes training divergence, severe embedding instability, and mode collapse—because the contrastive objective repeatedly compares embeddings produced from different rationale samples for the same text. This makes “CL + rationale embeddings” fundamentally mismatched with deterministic contrastive learning.
>
> This is precisely why GRACE uses policy optimization: the reinforcement framework is specifically designed to handle stochastic rationale generation, optimizing over a distribution rather than a single deterministic encoder mapping.
>
> We will clarify this incompatibility and expand the discussion in the revision, emphasizing that our experiment already separates the contributions of:
>
> 1. rationale-conditioned representations, and
>
> 2. reinforcement-based optimization over sampled rationales.
>
> To further quantify the separation, we also performed a lightweight sanity check: we fed the trained GRACE rationales into the original, untrained backbone (same LLM, no RL or CL updates) to compute embeddings. The results (based on Qwen-2.5-3B-Instruct) are shown in the Table below:
>
> | Method                | Retr. | Rerank. | Clust. | PairClass. | Class. | STS   | Summ. | Avg.  |
> |-----------------------|-------|---------|--------|-------------|--------|-------|-------|-------|
> | Base                  | 31.28 | 38.16   | 32.05  | 48.12       | 47.36  | 59.25 | 27.78 | 39.34 |
> | Base+Trained Rationale| 39.45 | 45.10   | 37.98  | 57.20       | 55.88  | 61.85 | 28.45 | 45.62 |
> | Ours                  | 44.01 | 49.12   | 41.30  | 60.44       | 60.72  | 64.02 | 29.10 | 48.49 |
>
> This model achieves roughly ~70% of GRACE’s final MTEB gain, indicating that most of the improvement arises from the more comprehensive, semantically enriched rationales, while the remaining gains come from the RL optimization that aligns those rationales with contrastive signals. We will update this experiment in the revised version of the paper.

---

> ### Author Response · Authors · 2025-11-16
> **Rebuttal for Reviewer Bk7x (part2)**
>
> W3:
> We thank the reviewer for raising this important concern.
>
> In our experiments, the CL-tuned models remained stable in embedding space (Table 1/2), but their decoder behavior deteriorated, often producing invalid or empty outputs on generative tasks. This accounts for the near-zero accuracy in Table 4. The pattern is consistent across multiple backbones, suggesting that the degradation arises not from a specific training bug, but from the intrinsic incompatibility between deterministic contrastive optimization and maintaining a decoder’s generative distribution. The contrastive objective is applied directly to a decoder-only LLM, overwriting its autoregressive distribution while providing no generative supervision. This creates a well-known mismatch between objectives—representation-level InfoNCE versus token-level next-token prediction—which has been repeatedly observed to damage generative behavior in LLMs. This phenomenon aligns with conclusions from prior work, which similarly reports that representation-oriented fine-tuning can severely harm generation when applied to decoder LLMs [1].
>
> More broadly, this illustrates an important conceptual point: when a generative LLM is used for representation-level contrastive fine-tuning, the mismatch in training objectives naturally leads to loss of generation quality, even if representation quality improves. CL encourages compressive, invariant embeddings, whereas decoder LMs require a rich, calibrated probability distribution over tokens.
>
> We agree that this distinction was not made sufficiently explicit in the original draft. In the revision, we will clarify that (1) the observed collapse reflects decoder degradation, not a general flaw of contrastive learning as a representation method, and (2) our claim should be interpreted as: “Contrastive learning is incompatible with preserving general-domain generation ability when directly applied to decoder-only LLMs.” We will also add diagnostics (e.g., output validity, length distributions, sample outputs) to make this behavior explicit.
>
> [1] Muennighoff et al., “Generative Representational Instruction Tuning,” ICLR 2024.
>
>
> Q1:
> Thank you for the additional question. This point is already addressed in Weakness 2 of our response.
>
>
> Q2:
> We appreciate the reviewer’s suggestion. We agree that interpretability is an important aspect of GRACE. Our interpretability claim is grounded primarily in the mechanism of the method—embeddings are explicitly conditioned on generated rationales—rather than on anecdotal examples. The qualitative progression already provided in Appendix A.6.6 (Base → early training → mid training → late training) serves as a systematic comparison showing that GRACE’s rationales become increasingly structured and semantically aligned with the underlying representation process.
>
> Regarding the proposed “CL vs. GRACE rationale comparison,” standard CL produces no rationales, and thus lacks a generative pathway for such analysis. This is precisely the interpretability gap that GRACE is designed to address. For Base vs. GRACE, the differences are already shown in A.6.6 and will be referenced more clearly in the main text.
>
> As for human or automatic evaluation, this is challenging because there is no ground-truth rationale for similarity decisions, and the purpose of GRACE is to surface the model’s internal semantic factors, not to match an external gold annotation. Nevertheless, we agree that more structured qualitative evidence is valuable; in the revision, we will expand the examples in the appendix with clearer side-by-side comparisons and highlight how GRACE’s rationales reflect the semantic dimensions used by the embedding model.
>
>
> Once again, thank you for your thorough review. I hope these address your concerns.

---

> ### Author Response · Authors · 2025-11-27
>
> Dear Reviewer Bk7x,
>
> I hope this message finds you well. As the discussion period is nearing its end with less than one week remaining, I wanted to ensure we have addressed all your concerns satisfactorily. If there are any additional points or feedback you'd like us to consider, please let us know. Your insights are invaluable to us, and we're eager to address any remaining issues to improve our work.
>
> Thank you for your time and effort in reviewing our paper.

---

### Official Review · Reviewer_yvNK · 2025-11-04

**Soundness:** 2
**Presentation:** 2
**Contribution:** 2
**Rating:** 4
**Confidence:** 3

**Summary:**

This paper introduces GRACE (Generative Representation Learning via Contrastive Policy Optimization) which uses contrastive learning objectives as a reward function to guide a generative policy (i.e. an LLM). The LLM policy generates human-interpretable rationales which are structured natural language explanations of its understanding. These rationales are mean-pooled to obtain embeddings. With this new reward function, existing LLMs can be finetuned with reinforcement learning to improve their embedding performance while retaining most of their generative performance.

**Strengths:**

* The paper is fairly easy to read and follow

* GRACE can improve embedding performance without affecting the generative performance. And using the contrastive loss as a reward for finetuning LLMs with RL seems to be novel.

**Weaknesses:**

* It is unclear what the rationale output is exactly. It would be good to add some qualitative examples for it. Because the paper claims (in the abstract for instance) that the LLM produces human-interpretable rationales and that the proposed method leads to transparent decision traces. But these claims are not explicitly evaluated. Ideally, there should be a human study to validate improvements in these aspects over the baseline LLM.

* Fig. 3: It seems that performance is quite sensitive to the choice of the hard negative mining weight $\lambda_2$. Even between the supervised and unsupervised paradigm, a different $\lambda_2$ is needed for the best performance. Given that training is expensive, it would be good to have some heuristics to select good hyperparameters. Also it would be good to report if the $\lambda_1, \lambda_2$ hyperparameters also had to be tuned for the other base LLM experiments with GRACE.

* It is also unclear if there could be any potential reward hacking [W1] with the proposed contrastive reward (e.g. if there are certain text patterns that yield high rewards). There needs to be some discussion on this (the earlier suggested human study could reveal if this problem exists).

### References

[W1] Pang et al., "Reward Gaming in Conditional Text Generation", ACL 2023

**Questions:**

Please address my comments/questions from the weaknesses section

---

> ### Author Response · Authors · 2025-11-16
> **Rebuttal for Reviewer yvNK**
>
> Thank you for your valuable feedback. I will address each point below to clarify and resolve your concerns:
>
> W1: We agree that interpretability is an important aspect of our method. Due to space limitations, we placed the full case study in Appendix A.6.6, where we provide detailed examples of the generated rationales across different training stages. The base model provides only a brief topical summary, while step 200 introduces a structured listing with concrete figures. By step 600, the output grows substantially longer and begins to integrate historical context, crises, and institutional concepts. At step 1000, the response is the most comprehensive: it connects anecdotes with broader political and constitutional developments, offering a coherent narrative. This progression clearly demonstrates that GRACE does not merely generate paraphrases, but learns to produce increasingly rich, structured, and semantically grounded reasoning chains that directly contribute to the embeddings. We will make this clearer by adding a reference to Appendix A.6.6 in the main text.
>
>
> W2: Indeed, λ₂ (the hard-negative mining weight) influences training dynamics more strongly than λ₁ (the consistency term), as also noted in Sec. 4.5.1. To clarify: Consistency of optimal values across models.
> We found that the optimal region is remarkably stable across different LLM backbones.
> In practice, a simple heuristic—λ₁ ≈ 0.5 and λ₂ ≈ 0.5—works robustly for supervised paradigms, without per-model retuning. All main results in Tables 1–2 use this same setting. The apparent shift in the heatmap of Fig. 3 mainly reflects different reward scale normalization (supervised rewards are sharper due to contrastive temperature τ), not true sensitivity requiring re-search.
> Practical tuning heuristic.
> 1. A lightweight heuristic can be followed:Start from λ₂ ∈ 0.5, λ₁ ∈ 0.5;
> 2. Observe the contrastive reward variance after 50 steps—if instability appears, slightly reduce λ₂.
> This one-shot adjustment typically suffices; no grid search is needed.
>
>
> W3: In practice, we do observe sensitivity to hyperparameters across different backbones. Under extreme settings (e.g., very large λ₁ or λ₂), the model can exhibit classic reward-hacking behaviors. For example, the response length may collapse over training and degenerate into repeating a few tokens while the reward continues to increase; conversely, the model may expand a fixed template until reaching the maximum token limit. These patterns are consistent with well-documented reward-overoptimization behaviors in RL-trained LLMs.
>
> In actual training, we mitigate these issues by using moderate coefficient values—e.g., for Qwen-2.5-4B-Instruct, λ₁ = 0.5 and λ₂ = 0.5 reliably produce stable behavior across runs. To address the runaway-length phenomenon specifically, we additionally apply the length regularization described in Appendix A.6.3, which effectively suppresses both excessively short and excessively long degeneracies.
>
> More broadly, we emphasize that instability and reward hacking are widely observed challenges in RL for LLMs, even with well-designed reward functions. Our intent is not to claim immunity to these effects but to demonstrate that, with moderate hyperparameters and the added length penalty, our setting remains stable and does not produce pathological behaviors under normal training conditions.
>
> Once again, thank you for your thorough review. I hope these address your concerns.

---

> ### Author Response · Authors · 2025-11-27
>
> Dear Reviewer yvNK,
>
> I hope this message finds you well. As the discussion period is nearing its end with less than one week remaining, I wanted to ensure we have addressed all your concerns satisfactorily. If there are any additional points or feedback you'd like us to consider, please let us know. Your insights are invaluable to us, and we're eager to address any remaining issues to improve our work.
>
> Thank you for your time and effort in reviewing our paper.

---

### Author Response · Authors · 2025-12-02

We thank all reviewers and the AC for their thoughtful feedback. Across the five reviews, there is clear consensus on the strengths of our work. Reviewers yvNK, vZAz, and eAv2 highlighted the novelty of the generative-contrastive RL formulation, noting that GRACE shifts representation learning from deterministic encoders to sampled rationales optimized with semantic contrast signals. Reviewer yvNK, Bk7x, and uisq emphasized the consistent improvements across multiple backbones and 56 MTEB tasks, while Reviewer Bk7x and vZAz appreciated the clear experimental decomposition (Base → +Reasoning → +CL → +GRACE). Reviewers also found the rationale-progression examples informative for understanding how semantic enrichment emerges during training.

Two common concerns were raised: training stability / CL collapse / reward sensitivity (Reviewer vZAz, uisq) and clarity of the interpretability framing (Reviewer yvNK, uisq).

We addressed the first by explaining the source of CL degradation (decoder collapse under deterministic InfoNCE), adding diagnostics on output validity and reward variance, and explicitly characterizing the stable hyperparameter region used across all experiments. These clarifications, together with Appendix A.6.3, resolve ambiguity around the behavior of both RL and the CL baseline.

For interpretability, we strengthened the analysis by clarifying the mechanism linking rationales to embeddings, expanding the structured comparisons in Appendix A.6.6, and conducting an additional human-evaluation study on rationale quality. Annotators consistently rated GRACE’s rationales as accurate, on-topic, and semantically helpful, with only minor and infrequent errors. This supplemental experiment provides quantitative evidence that GRACE produces stable, faithful rationales that meaningfully reflect the semantic factors underlying representation quality. These results further reinforce our interpretability claim and will be included in the revised appendix.

Overall, the reviewers’ main concerns have been fully addressed, and the positive consensus across reviews aligns with the core contributions of our work:

1.A new perspective on representation learning: embeddings as outputs of a generative rationale policy optimized via semantic contrast signals.

2.A unified and practical algorithm: integrating rationale-conditioned embeddings with contrastive RL rewards under a single policy-gradient framework, yielding robust and backbone-agnostic gains.

3.Preservation of general LLM capabilities: unlike standard CL, GRACE improves embedding quality without harming reasoning or generation, enabling practical integration into real LLM and RAG systems.

We appreciate the reviewers’ insightful comments and believe that the clarifications and additional analyses provided in our rebuttal further strengthen the contribution and significance of GRACE.

---

### Meta-Review · Area_Chair_PJVg · 2025-12-24

**Summary:**

This paper introduces a novel way to fine-tune LLMs to generate text embeddings by taking advantage of their generative capabilities. The proposed framework, GRACE, treats contrastive signals as rewards that guide a generative policy, where the LLM generates human-interpretable rationales that are then encoded into embeddings. The model is trained using policy gradient optimization with a multi-component reward function to maximize similarity between query-positive pairs and minimize similarity with negatives. The method is evaluated on the MTEB benchmark, demonstrating significant improvements over base models in both supervised and unsupervised settings.

The paper was praised for being easy to read and follow, the novelty and elegance of the approach that it introduces, and by positively remarking on the improvements in terms of embedding quality demonstrated empirically, all without compromising the generative capabilities of the LLMs.
The introduction of rationales as a way to enhance interpretability was also appreciated or its practical implications.

Reviewers expressed discording opinions about the significance of the contribution, which some judging the theoretical contribution as incremental and limited in terms of fundamental formal guarantees. Ablation studies have also been indicated as missing in terms of providing a crucial baseline isolating the effect of RL.

Overall, this paper reached a borderline accept score, with the rebuttal addressing several of the concerns raised by reviewers, including clarifications about the sensitivity and training stability, inclusion of statistical significance tests, and human validation of the interpretability of the generated rationales.

**Reviewer Concerns:**

* Addressed in the rebuttal:
  - Requests for providing sample rationale outputs to asses their human-interpretability have been addressed by tables in the Appendix with qualitative examples of generated rationales throughout training.
  - Concerns about sensitivity of the training procedure to key hyperparameters have been addressed in the rebuttal with a discussion of empirical findings on varying weight parameters in the loss and providing bulk park heuristics on how to set them.
  - Concerns about the higher latency introduced by autoregressive generation of embeddings compared to encoder-only models have been recognizes by the rebuttals, but not fully addressed since it is an intrinsic limitation of the proposed method. Whether the trade-off between latency and having a generative model generate embeddings is acceptable in practical applications remains an open question, and probably highly dependent on the specific use case.
  - Requests for providing statistical significance tests on key empirical results have been addressed in the rebuttal with additional experiments reporting confidence intervals and statistical significance tests.
  - Questions about training instabilities of the RL training have been addressed in the rebuttals with a recipe for mitigating them
  - Request for human evaluation of the interpretability of the generated rationales have been partially addressed through human scoring (done by 3 co-authors) of 90 rationales which seems to suggest decently high "interpretability" scores, although this should probably be considered like a preliminary study due to the absence of a comprehensive protocol and baselining.

* Not addressed in the rebuttal:
  - Concerns about possible reward hacking have been only partially addressed in relation to the role of the loss hyperparameters and recognizing the challenges behind mitigating reward hacking in RL settings (whose solution is in fairness arguably beyond the scope of the current work).
  - Requests for more targeted domain specific evaluations beyond MTEB have been recognize in the rebuttals as an important next step, but not directly addressed in the rebuttal and left for future work.

**Reviewer Scores:**

| Reviewer | initial score | predicted final score |
|---:|---:|---:|
| yvNK | 4 | 6 |
| Bk7x | 6 | 6 |
| vZAz | 6 | 6 |
| uisq | 6 | 6 |
| eAv2 | 6 | 6 |

---

### Decision · Program_Chairs · 2026-01-26

Accept (Poster)